# Thunderstorm Environments in Europe

Deborah Morgenstern[1, 2], Isabell Stucke[1, 2], Georg J. Mayr[1], Achim Zeileis[2], and Thorsten Simon[2, 3]

[1]Department of Atmospheric and Cryospheric Sciences (ACINN), University of Innsbruck, Innsbruck, Austria
[2]Department of Statistics, University of Innsbruck, Innsbruck, Austria
[3]Department of Mathematics, University of Innsbruck, Innsbruck, Austria

**Correspondence:** Deborah Morgenstern (deborah.morgenstern@uibk.ac.at)

**Abstract.** Meteorological environments favorable for thunderstorms are studied across Europe, including rare thunderstorm conditions from seasons with climatologically few thunderstorms. Using cluster analysis on ERA5 reanalysis data and EU-CLID lightning data, two major thunderstorm environments are found: Wind-field thunderstorms, characterized by increased wind speeds, high shear, strong large-scale vertical velocities, and low CAPE values compared to other thunderstorms in the same region; and mass-field thunderstorms, characterized by large CAPE values, high dew point temperatures, and elevated isotherm heights. Wind-field thunderstorms occur mainly in winter and more over the seas while mass-field thunderstorms occur more frequently in summer and over the European mainland. Several sub-environments of these two major thunderstorm environments exist.

Principal component analysis is used to identify four topographically distinct regions in Europe that share similar thunderstorm characteristics: The mediterranean, alpine-central, continental, and coastal regions, respectively. Based on these results it is possible to differentiate lightning conditions in different seasons from coarse reanalysis data without a static threshold or a seasonal criterion.

## 1 Introduction

Lightning, the defining characteristic of thunderstorms, can originate in a variety of meteorological settings. Some conditions that lead to lightning occur more frequently and are better understood than others. Numerous lightning climatologies are available but many focus on the dominant characteristics and seasons while infrequent thunderstorm conditions are often neglected. Thunderstorms during the cold season are generally rare but pose a serious threat to wind turbines and other tall structures because it has been observed that lightning strikes to tall infrastructure have no or only a weak annual cycle whereas lightning in general has a pronounced annual cycle (Stucke et al., 2022; Matsui et al., 2020; Vogel et al., 2016). This study describes thunderstorm environments occurring in Europe using a balanced view on all four seasons to include also seasonally infrequent thunderstorm conditions. A comparison between different regions provides a comprehensive overview of the lightning characteristics in Europe.

The general lightning pattern in Europe is well described in various climatologies (e.g., Taszarek et al., 2019; Enno et al., 2020; Poelman et al., 2016; Wapler, 2013; Taszarek et al., 2020a, b; Mäkelä et al., 2014; Vogel et al., 2016; Ukkonen and Mäkelä, 2019; Simon et al., 2017; Kotroni and Lagouvardos, 2016; Piper and Kunz, 2017; Anderson and Klugmann, 2014;

Hayward et al., 2022; Holt et al., 2001; Enno et al., 2013; Poelman, 2014; Taszarek et al., 2015; Schulz et al., 2005; Coquillat et al., 2022; Manzato et al., 2022; Simon and Mayr, 2022). There is a north-south gradient of lightning frequency with a maximum in northern Italy and the Mediterranean. Lightning in central Europe follows a clear annual cycle with a maximum over land in summer (MJJA) and a secondary peak in fall and early winter (SONDJ) in the Mediterranean (Taszarek et al., 2019; Poelman et al., 2016; Enno et al., 2020). In south-central Europe, the annual cycle is less pronounced and has sometimes two lightning maxima along with a local minimum in summer (Taszarek et al., 2019). There are differences in the annual lightning cycle: Offshore and coastal areas have a lower amplitude and a later maximum compared to inland or mountainous locations (Wapler, 2013; Enno et al., 2013). In the northern Atlantic region, occasional intense thunderstorms are possible, even though the climatological thunderstorm activity is low (Enno et al., 2020). There, lightning in the cold season (Oct.-Apr.) occurs predominantly over the seas (North Sea, Baltic Sea, Atlantic) and less so over the land (Mäkelä et al., 2014). The question remains whether there are meteorologically different thunderstorm conditions at work, resulting in these spatial and temporal differences in lightning characteristics across Europe.

Many processes influencing lightning occurrence are known: The diurnal lightning cycle in Europe peaks in the afternoon over land and at night over the sea (Taszarek et al., 2020a; Enno et al., 2020; Manzato et al., 2022). Nighttime offshore lightning (Bay of Biscay, the North Sea, and the Baltic Sea) is explained by convection initiated over land and advected out to sea, where lightning activity endures longer as the sea surface temperatures are unaffected by nighttime cooling (Enno et al., 2020). The most pronounced diurnal cycle is found over mountainous areas and commonly explained by the topography. Complex terrain favors more unstable environments (less CIN when the surface is close to the level of free convection), mechanical forcing (forced lifting), and thermal forcing (elevated heating leads to positive buoyancy and up-mountain flow; Manzato et al., 2022). This is particularly relevant after the snow has melted at higher elevations (Simon and Mayr, 2022). Most of continental Europe experiences $20 - 40$ thunderstorm days annually, but the mountain ranges in southern Europe have thunderstorm frequencies of $> 60$ thunderstorm days per year (e.g., northern Italy).

The sea has an effect on lightning as the number of lightning strokes and the sea-surface temperature are positively correlated in fall (Kotroni and Lagouvardos, 2016). Mallick et al. (2022) even suggests the use of sea surface temperature as a proxy for seasonal lightning forecasts. Warm oceanic currents are known to increase lightning densities in each season and particularly so in winter (Iwasaki, 2014; Holle et al., 2016). Wintertime lightning occurs usually in mid-latitudinal cyclones (Bentley et al., 2019) and lightning bands are found in wintertime storm tracks (Zhang et al., 2018; Virts et al., 2013). In general, the European lightning patterns are well described (e.g., Wapler and James, 2015; Enno et al., 2014), but the meteorological drivers leading to lightning in the winter compared to summer are less understood. High structures such as wind turbines or radio towers increase the occurrence of lightning (March et al., 2016), especially in the cold season (Vogel et al., 2016; Pineda et al., 2018) so that lightning damage to infrastructure is evenly distributed over the year even though lightning occurrence in the surroundings has a strong annual cycle (Stucke et al., 2022).

The 2018 update of the lightning protection standard for wind turbines introduces different lightning threats in winter and in summer (Méndez et al., 2018; IEC 61400-24, 2019). Using the maps from March et al. (2016), the environmental factor in the standard now includes the local threat of winter lightning. While considering winter lightning is a good first step, the

quality of the risk assessment could be improved because the maps are very coarse, underestimate winter lightning, and use a static threshold ($< 5\,°C$ at $900\,hPa$). The reasons for the insufficient consideration of lightning in winter in the standard are the different processes leading to upward lightning and the limited meteorological knowledge of lightning in the cold season (Becerra et al., 2018).

One approach to investigate these differences would be to numerically simulate individual thunderstorms, which may require horizontal resolutions of $\mathcal{O}(100\,m)$ (Bryan et al., 2003) and still fail to make thunderstorms appear at the correct times and places. Our approach takes advantage of already knowing where and when lightning occurs from measurements. Thunderstorm environments can then be identified from reanalysis data that do not need to explicitly resolve thunderstorms or the processes leading to electrification. Morgenstern et al. (2022) used this approach and found that thunderstorms in the cold season differ physically from thunderstorms in the warm season in northern Germany. They describe wind-field thunderstorms dominant in winter in contrast to CAPE (mass-field) thunderstorms typical for summer. Here, their findings are extended to large parts of Europe to answer two questions:

– Are there regions in Europe, where thunderstorms occur under similar meteorological conditions?

– What characterizes thunderstorms in different meteorological environments and how do they vary seasonally across Europe?

To answer these questions, Europe is divided into 12 domains (Sect. 2). Applying similar methods as in Morgenstern et al. (2022), principal component analysis finds the answer to the first question, and $k$-means clustering to the second (Sect. 3). The general thunderstorm conditions in Europe are presented in Sect. 4.1 revealing that the 12 domains can be summarized into four regions with similar lightning characteristics. Cluster analysis on each domain in Sect. 4.2 leads to two main thunderstorm environments and three variations thereof. These thunderstorm environments are then analyzed seasonally (Sect. 4.3) and compared to one another (Sect. 4.4). The results are discussed in Sect. 5, and Sect. 6 summarizes the main findings.

## 2 Data

Two data sets are incorporated in this study (cf. Morgenstern et al., 2022): meteorological reanalysis data (ERA5, Sect. 2.1) representing meteorological environments, and lightning observations (EUCLID, Sect. 2.2).

### 2.1 Meteorological data: ERA5

Meteorological data are extracted from the single-level and model-level data of the ERA5 global reanalysis provided by ECMWF (Hersbach et al., 2020). The distance between vertical model levels varies from $10\,m$ near the ground to $320\,m$ in the lower stratosphere (lowest 74 levels). The horizontal resolution is $0.25\,°$ latitude–longitude and the temporal resolution is one hour. A binary land-sea mask sets ERA5 grid cells with at least $35\,\%$ land to land to capture the influence of sub-grid islands.

ERA5 provides consistent data on the state of the atmosphere at a scale larger than individual thunderstorms. It directly contains and allows to derive additional variables related to various atmospheric processes relevant to lightning: The presence of differently sized cloud particles and strong motions leading to collision and subsequent separation of the particles. Almost one hundred such variables were computed and exploratively analyzed. By eliminating highly correlated variables that provide limited additional information, a set of 25 variables remains (Table 2). The set is indicative of substantial clouds (e.g., moisture, large-scale vertical velocity, precipitation, cloud size), charge transfer (e.g., ice, snow, supercooled liquids), and charge separation (e.g., shear, wind, CAPE, CIN). As some required variables are not directly available at ERA5 (https://www.doi.org/10.24381/cds.adbb2d47, accessed 2023-02-15), they are derived using also the model level data. These additionally derived variables include variables such as the height of the $-10\,°C$ isotherm, cloud mass between $-10$ and $-40\,°C$ (ice and snow), and the product of maximum large-scale vertical velocity and liquid particles between $-8$ and $-12\,°C$ (*vertical liquids flux*). Further derived variables are cloud size, cloud shear, wind speed at cloud base, maximum upward vertical velocity, and the temperature difference between the air mass at $1000\,m$ a.g.l. and the surface (sea surface temperature or skin temperature). All 25 variables are listed in Table 1 and details about them are provided in the online supplement (Morgenstern et al., 2023).

To ease interpretation, physical-based categories group the variables: *Mass-field variables* refer to temperature, pressure, and humidity. *Surface-exchange variables* include atmospheric fluxes interacting with the surface. *Wind-field variables* cover everything related to wind. *Cloud-physics variables* refer to measures directly related to clouds. *Topographic variables* consist of the surface geopotential height (orography) and a binary land-sea mask.

## 2.2 EUCLID lightning data and geographical domains

Lightning data are provided by the European Cooperation for Lightning Detection (EUCLID Schulz et al., 2016; Poelman et al., 2016), a cooperation of several local lightning location systems (LLS) in Europe. Only cloud-to-ground lightning flashes between 2010–2020 are considered, as this period is most stable regarding the hardware and software configuration of the network. If at least one lightning flash occurred within an ERA5 cell in a given hour, the whole cell-hour is regarded as one lightning observation.

The EUCLID territory is separated into 12 domains with rather homogeneous topography and lightning detection efficiency (Fig. 1) aiming to represent typical European landscapes. Domain A covers large parts of the North Sea including the surrounding coastlines. It marks the furthest northern EUCLID domain with sufficient lightning detection efficiency and sufficient lightning observations in each season and year. Domain B covers large parts of Denmark and northern Germany as well as parts of the North Sea and the Baltic Sea. It also covers parts of the domain analyzed in Morgenstern et al. (2022) to which the current study is an extension. Domain C is representative of the southern Baltic Sea and Poland, except for the Carpathian mountains in the South. Domain D covers the Gulf of Biscay, an Atlantic domain. Domain E covers the whole Iberian Peninsula including the Pyrenees and the surrounding coastal areas and is characterized by highlands. Domain F covers large parts of France and Belgium, being a topographically less homogeneous domain. Domain G covers hills in Germany, Czech Republic, Southern Poland, and Slovakia. Domain H covers the east-west elongated part of the European Alps. Domain I covers Hungary, Croatia,

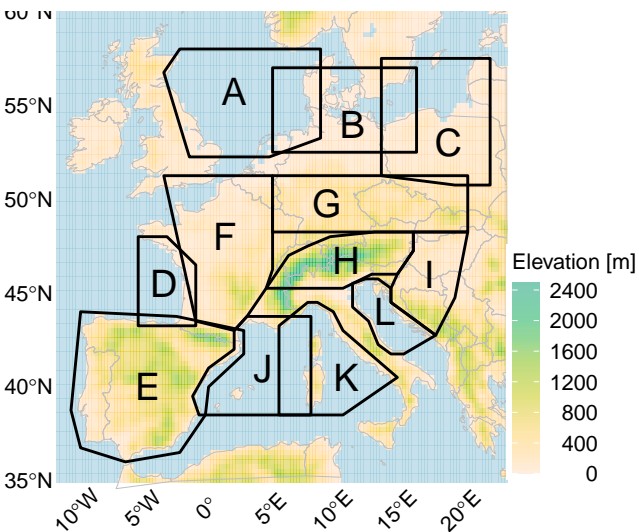

**Figure 1.** Overview of domains. Topographic data is based on ERA5 orography with a resolution of $0.25\,^\circ \times 0.25\,^\circ$ longitude/latitude. A land-sea mask is applied where each cell containing $> 35\,\%$ land is regarded as land.

and Bosnia and Herzegovina, a basin surrounded by mountain ranges. Domain J covers the Balearic islands in the northwestern Mediterranean Sea and surrounding coastlines. Domain K covers the Tyrrhenian Sea and the Islands of Corsica and Sardinia and Italian coastal areas. Finally, domain L covers the northern part of the Adriatic Sea including the surrounding coastlines.

## 3 Methods

To investigate spatio-temporal lightning characteristics, lightning data sets for the 12 domains are constructed that have the same number of observations from each season. The lightning data sets are then combined with 25 ERA5 variables representing the atmospheric conditions at the hour of the lightning observations. Using the domain means, a *spatial* lightning analysis for Europe is performed with the help of a principal component analysis. Then, thunderstorm *environments* are found individually on each domain by a cluster analysis with $k = 3$ clusters. A *seasonal* lightning analysis follows by analyzing how many observations from each season have been classified into which thunderstorm environment. Finally, the thunderstorm environments are compared to one another using again a principal component analysis.

### 3.1 Composition of data

EUCLID lightning data is aggregated to the spatio-temporal resolution of ERA5 resulting in binary cell-hours indicative of lightning. For each lightning cell-hour, ERA5 data at the respective cell and from the last full hour is taken to capture the build-up of the thunderstorms. Accumulated variables, such as precipitation, are taken from the next full hour to capture everything within the hour in which lightning was observed. Only cell-hours with lightning are considered. To investigate

seasonal differences, the available data are reduced to contain the same number of lightning cell-hours from each season (winter = DJF, spring = MAM, summer = JJA, fall = SON). Therefore a random sample without replacement is drawn from the seasons with more lightning cell-hours. Depending on the domain size and general lightning frequency, the data set in each domain consist of $5\,320 - 40\,000$ observations (Table 1). For robustness, the whole analysis is performed on $50$ different samples in each domain. A visual comparison of the resulting figures reveals qualitatively the same results between these repetitions. Hence, the samples are representative and it is sufficient to discuss only one sample in the following.

$k$-means clustering requires scaled input variables that follow rather similar distributions. Therefore all ERA5 variables are square root transformed and scaled to a mean of zero and a standard deviation of one.

$$x_\mathrm{t} = \mathrm{sign}(x)\sqrt{\mathrm{abs}(x)}, \tag{1}$$

with $x$ being the original ERA5 value and $x_\mathrm{t}$ its transformation.

$$x_\mathrm{s} = \frac{(x_\mathrm{t} - \mu)}{\sigma}, \tag{2}$$

$\mu$ and $\sigma$ are the empirical mean and standard deviation and $x_\mathrm{s}$ is the scaled value. The applied algorithm is supplied in the online supplement (Morgenstern et al., 2023). For the cluster analysis in Sect. 4.2, transformation and scaling are performed individually on each domain. For the domain comparison in Sect. 4.1 (Sect. 4.4), scaling is performed on the domain means (cluster means) of all domains together.

### 3.2 Statistical methods

Principal component analysis (Mardia et al., 1995) is an approach for dimension reduction that computes several linear combinations of projected input data (principal components, PC) aiming to capture as much variability from the data as possible. The first PC explains the most variance and each following PC is oriented perpendicular to the previous PC explaining less and less variance. Omitting the later PC's results in the intended dimension reduction. In this study, the first two PC's are used as axes for a so-called biplot to visualize the variance in the 25-dimensional data.

$k$-means cluster analysis (MacQueen, 1967) is a data-driven approach to find groups in data, aiming at maximum similarity within and minimum similarity between the groups. The similarity is measured with the squared euclidean distance between each observation and the cluster means. Starting with $k$ random cluster means, new cluster means are calculated iteratively to which the observations are assigned forming the clusters. The optimal number of clusters $k$ for the data used in this study is derived from the sum of the squared residuals and ranges from 2 to 4. The results for $k = 3$ are presented in detail, and the results for $k = 2$ and $\geq 4$ are also described. Cluster analysis is used to identify different thunderstorm environments. To account for possible regional differences, clustering is performed separately on each of the 12 topographically homogeneous domains.

The online supplement provides the R code to replicate the cluster analysis and the principal component analysis (Morgenstern et al., 2023).

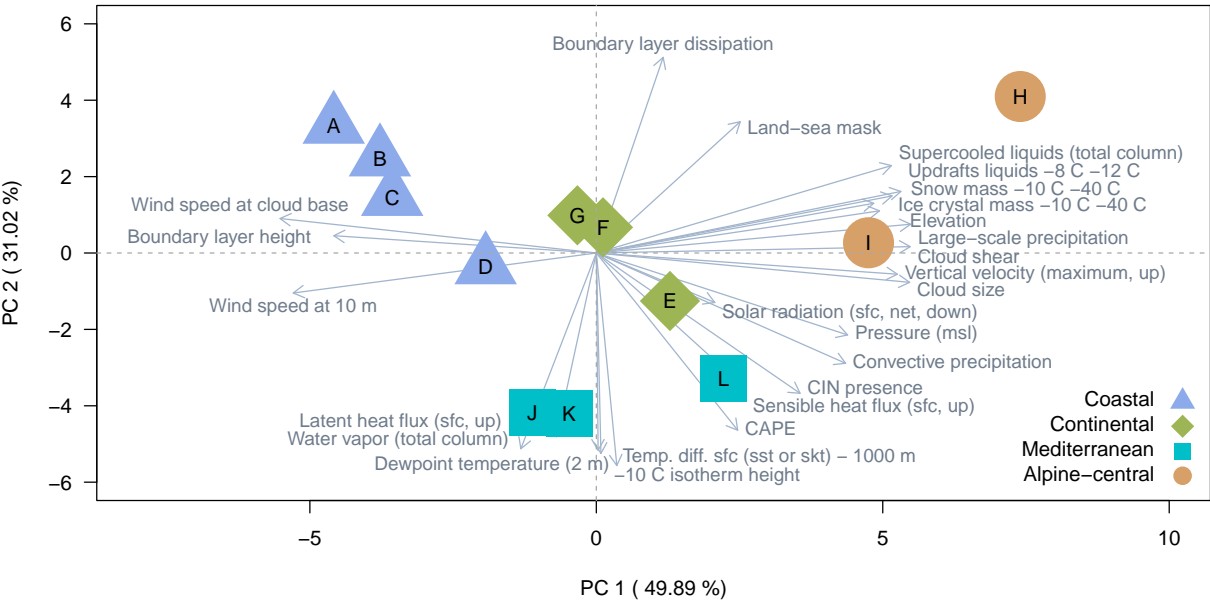

**Figure 2.** Spatial lightning differences in Europe. Based on Table 1, a PCA is computed and displayed as a biplot using the first two principal components (PC) as axes. Labeled arrows (loadings) indicate the contribution of each meteorological input variable to the variance in the respective direction. Domains with similar characteristics are indicated with the same color and symbol and labeled accordingly (legend).

## 4    Results

Thunderstorms in 12 European domains are compared to one another using principal component analysis (PCA) investigating which domains have in general similar meteorological characteristics during lightning throughout the year (Sect. 4.1). Then
thunderstorm environments are found by $k$-means clustering and a decision tree is presented to differentiate them (Sect. 4.2). Finally, the thunderstorm environments are seasonally analyzed and compared in Sections 4.3 and 4.4.

### 4.1    Regional differences between thunderstorms in Europe

This section investigates how the meteorological thunderstorm conditions vary regionally in Europe and whether some of the 12 domains can be grouped together based on their meteorological similarities during lightning throughout the year.
Table 1 presents the meteorological mean values of 25 variables separately for each domain. These are typical values for thunderstorms throughout the year for the respective domains and are considered as 'baselines'. A principal component analysis (Fig. 2) makes it easier to spot differences and commonalities between these domains. The first two principal components (x-axis and y-axis) explain together about 80 % of the variance within the data. The further the domains (colored symbols) are

**Table 1.** Meteorological mean values during lightning for each domain. Domains with similar characteristics are grouped into four regions, for which a mean value is also given. ERA5 variables are grouped by their meteorological category.

| | CAPE | CIN presence [*] | Pressure (msl) | Water vapor (total column) | Temperature dew point (2 m) | -10 C isotherm height [*] | Temp. diff. sfc (sst or skt) - 1000 m a.g.l. [*] | Boundary layer height | Solar radiation (sfc, net, down) | Sensible heat flux (sfc, up) | Latent heat flux (sfc, up) | Wind speed at 10 m | Wind speed at cloud base [*] | Cloud shear [*] | Boundary layer dissipation | Vertical velocity (maximum, up) [*] | Vertical liquids flux -8 C -12 C [*] | Ice crystal mass -10 C -40 C [*] | Snow mass -10 C -40 C [*] | Supercooled liquids (total column) | Convective precipitation | Large-scale precipitation | Cloud size [*] | Land-sea mask [*] | Elevation | Nr. of observations |
|---|---|---|---|---|---|---|---|---|---|---|---|---|---|---|---|---|---|---|---|---|---|---|---|---|---|---|
| Unit | J kg⁻¹ | binary | hPa | kg m⁻² | °C | m a.g.l. | K | m | W m⁻² | W m⁻² | W m⁻² | m s⁻¹ | m s⁻¹ | m s⁻¹ | W m⁻² | Pa s⁻¹ | g Pa s⁻¹ | g m⁻² | g m⁻² | g m⁻² | mm | mm | m | binary | m | nr. |
| Category | Mass field | | | | | | | Surface exchange | | | | Wind speed | | | | | Cloud physics | | | | | | | Topography | | |
| **Coastal region** | | | | | | | | | | | | | | | | | | | | | | | | | | |
| A | 178 | 0.37 | 1004.4 | 20.5 | 9.4 | 3584 | 6.3 | 820 | 113 | 10 | 77 | 7.9 | 13.3 | 14.4 | 7 | 0.678 | 7 | 66 | 93 | 40 | 0.052 | 0.012 | 6791 | 0.39 | 7 | 30792 |
| B | 228 | 0.43 | 1005.8 | 21.7 | 9.9 | 3706 | 6.3 | 843 | 131 | 7 | 82 | 6.6 | 12.6 | 14.3 | 8 | 0.658 | 7 | 65 | 93 | 41 | 0.052 | 0.011 | 6994 | 0.67 | 27 | 27716 |
| C | 275 | 0.50 | 1007.5 | 23.4 | 10.6 | 3898 | 6.1 | 820 | 139 | 3 | 81 | 5.4 | 11.6 | 13.8 | 8 | 0.630 | 5 | 58 | 83 | 41 | 0.051 | 0.011 | 7010 | 0.83 | 92 | 5320 |
| D | 250 | 0.43 | 1008.3 | 23.6 | 11.7 | 3965 | 6.9 | 760 | 114 | 20 | 96 | 6.5 | 11.5 | 16.4 | 6 | 0.902 | 9 | 68 | 96 | 41 | 0.063 | 0.013 | 7072 | 0.49 | 67 | 40000 |
| Mean | 233 | 0.43 | 1006.5 | 22.3 | 10.4 | 3788 | 6.4 | 811 | 124 | 10 | 84 | 6.6 | 12.3 | 14.7 | 7 | 0.717 | 7 | 64 | 91 | 41 | 0.055 | 0.012 | 6967 | 0.60 | 48 | |
| **Continental region** | | | | | | | | | | | | | | | | | | | | | | | | | | |
| E | 299 | 0.52 | 1011.9 | 22.1 | 11.4 | 3927 | 8.9 | 861 | 185 | 49 | 90 | 4.2 | 8.2 | 17.6 | 6 | 0.941 | 11 | 64 | 94 | 52 | 0.065 | 0.018 | 7678 | 0.81 | 529 | 40000 |
| F | 284 | 0.50 | 1008.3 | 23.0 | 11.3 | 3918 | 6.8 | 805 | 148 | 15 | 87 | 4.5 | 10.4 | 17.1 | 7 | 0.841 | 11 | 72 | 112 | 48 | 0.062 | 0.018 | 7556 | 0.93 | 254 | 40000 |
| G | 300 | 0.53 | 1009.9 | 22.7 | 10.4 | 3851 | 6.9 | 839 | 168 | 13 | 91 | 3.9 | 9.7 | 16.0 | 9 | 0.723 | 9 | 68 | 105 | 45 | 0.055 | 0.015 | 7464 | 1.00 | 385 | 27016 |
| Mean | 294 | 0.52 | 1010.0 | 22.6 | 11.0 | 3899 | 7.5 | 835 | 167 | 26 | 89 | 4.2 | 9.4 | 16.9 | 7 | 0.835 | 10 | 68 | 104 | 48 | 0.061 | 0.017 | 7566 | 0.91 | 389 | |
| **Alpine-central region** | | | | | | | | | | | | | | | | | | | | | | | | | | |
| H | 315 | 0.53 | 1011.8 | 19.5 | 9.3 | 3538 | 5.9 | 472 | 135 | 18 | 72 | 1.9 | 5.1 | 21.6 | 10 | 1.223 | 45 | 85 | 160 | 90 | 0.066 | 0.047 | 8144 | 1.00 | 891 | 17040 |
| I | 352 | 0.61 | 1010.1 | 24.0 | 11.7 | 4043 | 6.3 | 602 | 155 | 22 | 80 | 2.4 | 6.4 | 18.8 | 8 | 1.006 | 25 | 79 | 141 | 67 | 0.071 | 0.032 | 8179 | 1.00 | 461 | 39224 |
| Mean | 334 | 0.57 | 1011.0 | 21.8 | 10.5 | 3791 | 6.1 | 537 | 145 | 20 | 76 | 2.2 | 5.8 | 20.2 | 9 | 1.115 | 35 | 82 | 151 | 79 | 0.069 | 0.040 | 8162 | 1.00 | 676 | |
| **Mediterranean region** | | | | | | | | | | | | | | | | | | | | | | | | | | |
| J | 436 | 0.57 | 1010.5 | 25.0 | 13.4 | 4178 | 8.7 | 699 | 134 | 27 | 106 | 6.3 | 9.6 | 16.3 | 3 | 0.811 | 9 | 65 | 101 | 39 | 0.060 | 0.018 | 7280 | 0.21 | 55 | 40000 |
| K | 495 | 0.60 | 1009.7 | 24.2 | 13.3 | 4111 | 8.9 | 685 | 151 | 30 | 105 | 5.5 | 8.4 | 15.4 | 3 | 0.840 | 8 | 65 | 93 | 40 | 0.067 | 0.016 | 7396 | 0.34 | 117 | 40000 |
| L | 524 | 0.63 | 1009.0 | 24.6 | 13.1 | 4096 | 8.4 | 576 | 133 | 23 | 91 | 4.6 | 7.5 | 18.1 | 3 | 0.984 | 14 | 75 | 113 | 49 | 0.074 | 0.022 | 7983 | 0.45 | 124 | 40000 |
| Mean | 485 | 0.60 | 1009.7 | 24.6 | 13.3 | 4128 | 8.7 | 653 | 139 | 27 | 101 | 5.5 | 8.5 | 16.6 | 3 | 0.878 | 10 | 68 | 102 | 43 | 0.067 | 0.019 | 7553 | 0.33 | 99 | |

[*] Derived variables.

[△] Accumulated variables considered at the next full hour after lightning observation. All other variables are considered at the last full hour.

CAPE: convective available potential energy, CIN: convective inhibition, msl: mean sea level, a.g.l.: above ground level,

Temp. diff. sfc (sst or skt) - 1000 m a.g.l.: Temperature difference between the surface and 1000 m a.g.l., where sfc is either sea surface temperature or skin temperature.

from the origin, the larger their contribution to the variance in the respective direction. Domains with similar meteorological characteristics are located in close proximity to one another within this diagram. The loadings (labeled arrows) indicate the direction and strength of individual meteorological variables responsible for the variation in the respective direction.

The domains A, B, C, and D (blue triangles) are all located in the top-left of Fig. 2. The labeled arrows indicate that these domains are physically characterized by increased boundary layer heights ($\sim 800\,\mathrm{m}$) and increased wind speeds at $10\,\mathrm{m}$ ($\sim 6\,\mathrm{m\,s^{-1}}$) and at cloud base ($\sim 12\,\mathrm{m\,s^{-1}}$) relative to all other domains (Table 1). Long arrows pointing in opposite directions of domains A–D indicate decreased values. For example, decreased values in CAPE, CIN, pressure, and cloud size. The temperature difference between the ocean (or skin temperature over land) and the air at $1000\,\mathrm{m}$ altitude is on average $6.4\,^\circ\mathrm{C}$ indicating a rather cool ground (Table 1). The regional characteristics of domains A–D are their large ocean areas including coastlines, hence they are grouped together as the 'coastal' region. The domains J, K, and L (turquois squares) gather in the lower part of Fig. 2. Their common physical characteristics relative to the other domains are high $2\,\mathrm{m}$ dew point temperatures above $13\,^\circ\mathrm{C}$, elevated $-10\,^\circ\mathrm{C}$ isotherm heights of more than $4100\,\mathrm{m}$, large CAPE values ($> 400\,\mathrm{J\,kg^{-1}}$), the presence of CIN, and high amounts of total column water vapor ($\sim 25\,\mathrm{kg\,m^{-2}}$). The temperature difference between the surface and at $1000\,\mathrm{m}$ altitude is more than $8\,\mathrm{K}$, indicating a warm ground (Table 1). Regionally, all these domains are located in the Mediterranean, and are hence grouped together as the 'mediterranean' region. The domains H and I are located on the right or top-right of Fig. 2 (orange circles) and are physically characterized by increased cloud-physics variables and increased wind-field variables such as various increased cloud particle concentrations (ice, snow, and liquids), increased cloud shear, increased large-scale vertical velocities, and large amounts of large-scale precipitation. Regionally, both domains are located in Central Europe and are influenced by mountainous topography. Hence, they are grouped together as the 'alpine-central' region. The remaining domains E, F, and G (green diamonds) are all located in the center of Fig. 2 and share the physical characteristics of mostly average values but increased surface exchange values. Spatially, their common characteristic is their location on the European mainland and hence they are grouped together as the 'continental' region.

Figure 3 shows how the meteorological conditions vary in detail between the four regions. It compares the mean meteorological values for each region (lines) relative to the others and shows how distinct thunderstorm conditions in Europe are. Scaled values (y-axis) close to zero indicate average values in the respective variable (x-axis) compared to the other regions. The figure shows that e.g. CAPE is in general much higher in the mediterranean region compared to the others, or that increased wind speed in the alpine-central region refers to much lower values than in all other regions. Appendix A1 dives deeper and presents the means of each domain separately.

With this, the spatially different thunderstorm conditions in Europe are described and summarized into four regions, each with shared physical characteristics during lightning throughout the year: the mediterranean region, the alpine-central region, the coastal region, and the continental region.

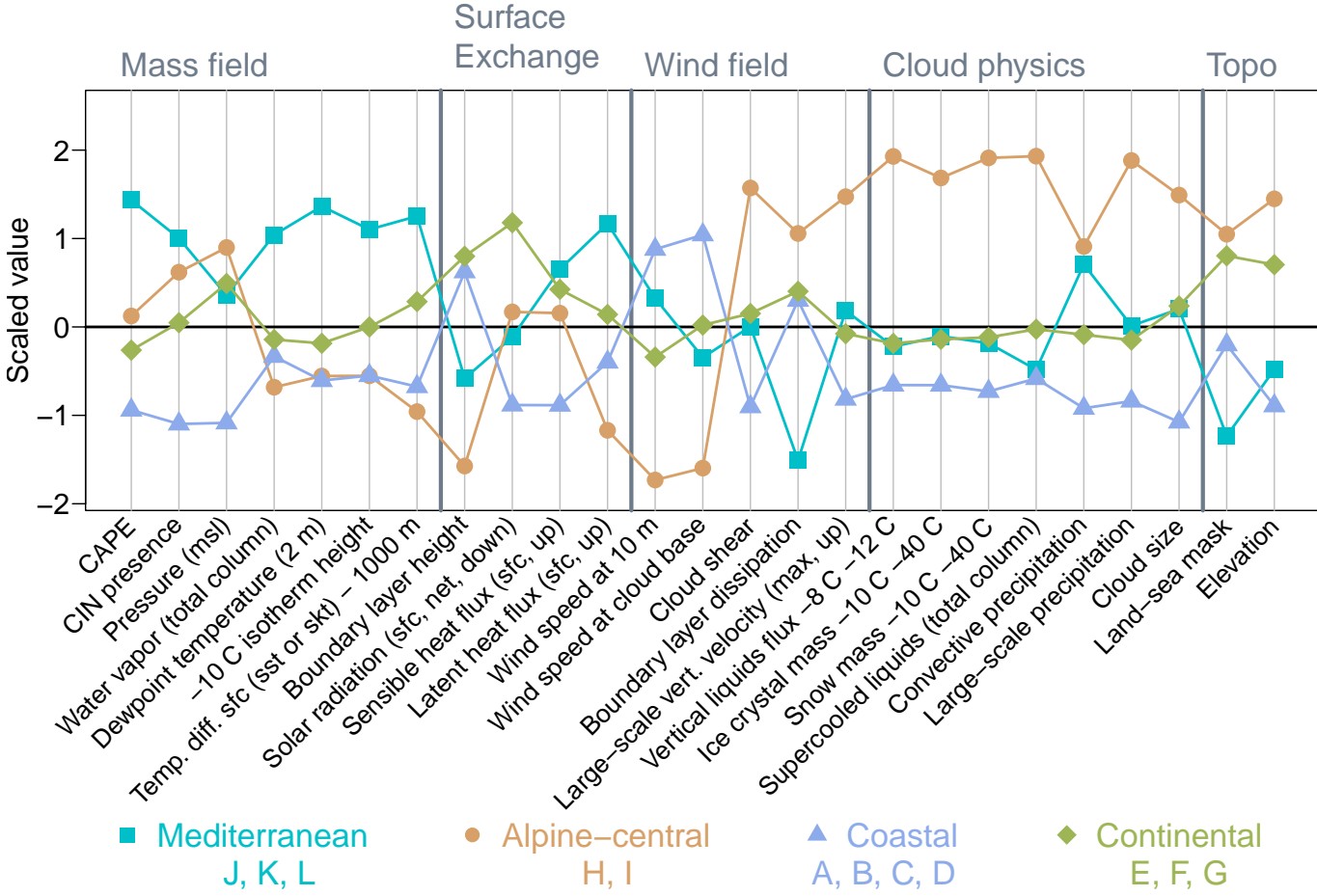

**Figure 3.** Parallel coordinate plot of mean meteorological values based on the means provided in Table 1. Similar domains are summarized into regions having the same color and symbol. ERA5 variables are scaled to mean zero and standard deviation one. More details are provided in Appendix A1.

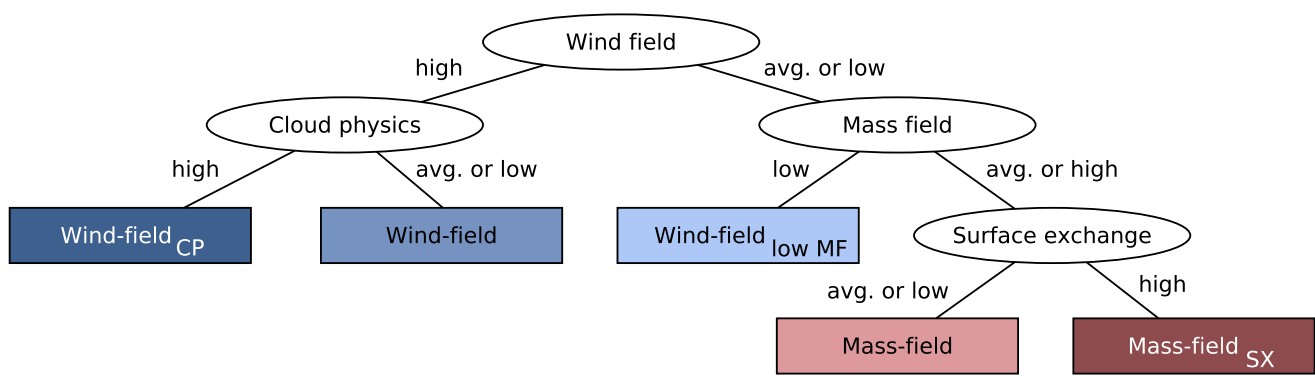

**Figure 4.** Decision tree to label the clusters to the two major thunderstorm environments and the three sub-environments (colored boxes). The abbreviations in the sub-environments stand for: CP = additionally increased cloud-physics variables, lowMF = low mass-field variables, and SX = increased surface-exchange variables compared to other thunderstorms in the same domain.

## 4.2 Thunderstorm environments

After finding four regions where thunderstorms have similar characteristics throughout the year, the next goal is to investigate whether *individual* thunderstorms occur under similar larger-scale meteorological conditions, i.e. whether different thunderstorm environments exist.

Cluster analysis with $k = 3$ is performed separately on every domain to find thunderstorm environments (clusters) relative to the overall lightning characteristics in that domain. Each found cluster from each domain is then described by its driving meteorological characteristics using the average values of the 25 input variables (cluster means). Then the average values within the physically-based categories (mass field, wind field, cloud physics, surface exchange, and topography) are computed for each cluster to yield an overall characterization. Two major thunderstorm environments emerge, as the wind-field category and the mass-field category always deviate substantially. The decision tree in Fig. 4 distinguishes between these two thunderstorm environments and helps to identify further sub-environments. *Wind-field thunderstorms* are characterized by increased wind-field values and rather low mass-field variables compared to other thunderstorm conditions in the same domain, and are indicated by bluish colors. There are two wind-field sub-environments: *Wind-field$_{CP}$ thunderstorms* (dark blue) that have additionally enhanced cloud-physics variables (CP), and *wind-field$_{lowMF}$ thunderstorms* (light blue), where the dominant feature is particularly low mass-field values compared to other thunderstorms (low MF), while wind-field variables and cloud-physics variables are at their average values. Compared to conditions without lightning, the moisture and temperature profiles (mass-field values) are still increased in wind-field environments (Morgenstern et al., 2022).

The other major thunderstorm environment, *mass-field thunderstorms*, is characterized by average or increased mass-field variables compared to the wind-field thunderstorms plus often decreased surface-exchange variables and is indicated by red-

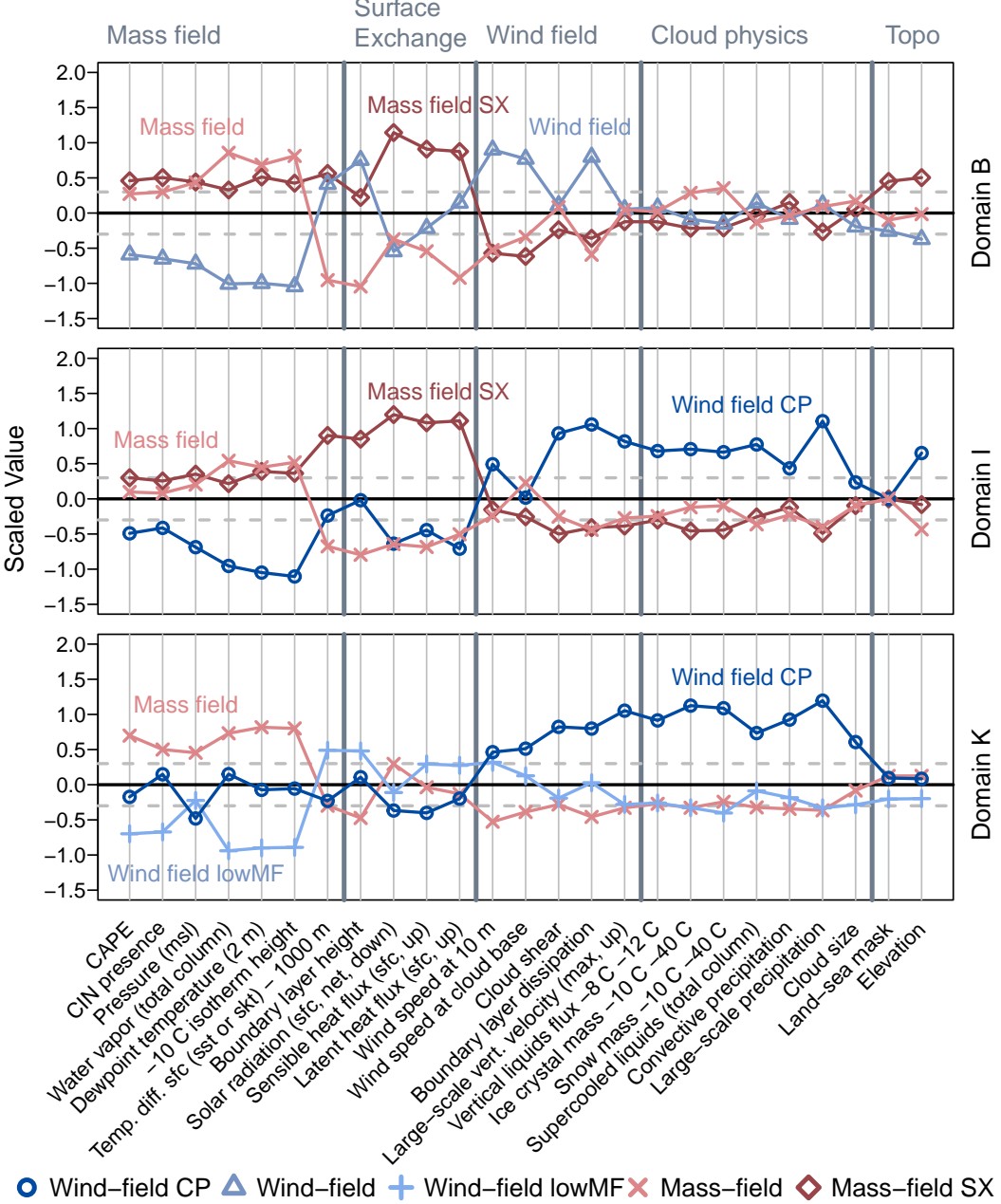

**Figure 5.** Parallel coordinate plot of thunderstorm environments (clusters) found in three representative domains (B, I, K; panels) and expressed by the cluster means (lines) for each meteorological variable (x-axis) using scaled values (y-axis). The ERA5 variables are grouped by their meteorological category (secondary x-axis). Colors indicate the thunderstorm environments (legend; blues = wind-field thunderstorms, reds = mass-field thunderstorms). The abbreviations in the sub-environments stand for: CP = additionally increased cloud-physics variables, lowMF = decreased mass-field variables, and SX = increased surface-exchange variables compared to other thunderstorms in the same domain. Category means between the dashed lines (+/- 0.3) are considered as average.

dish colors. There is one mass-field sub-environment, *mass-field$_{SX}$ thunderstorms* (dark red), with increased surface-exchange variables (SX) and sometimes average mass-field values.

A more detailed analysis of the thunderstorm environments is undertaken by investigating the cluster means of three representative domains in Fig. 5. The cluster means (lines) are displayed as scaled values (y-axis) of the meteorological variables (x-axis). A value close to zero indicates a typical value for thunderstorms in that domain while large deviations indicate large differences in this variable for different thunderstorm environments in that domain (as standard deviations). The average values or 'baselines' (y = 0) for each domain in Fig. 5 are provided in Table 1. Baselines are required to decide whether a value refers to a general high value. The unscaled cluster means are given in Table 2. The dashed lines indicate the +/- 0.3 threshold used for the decision tree (Fig. 4). Results for all domains are supplied in Appendix B1–B4, Table 2, and in the online supplement (Morgenstern et al., 2023). For robustness, each cluster analysis is repeated 50 times, but only one representative result is shown.

Figure 5 shows, that the wind-field thunderstorms (middle-blue triangles) in domain B occur together with enhanced wind speeds, enhanced boundary layer dissipation, lower $-10\,^\circ$C isotherm heights, little water vapor, small CAPE, and large boundary layer heights of more than $1200\,\mathrm{m}$ (Table 2) compared to other thunderstorms in domain B. Different from this, the wind-field$_{CP}$ thunderstorms (dark-blue circles) in domains I and K show thick clouds ($8731\,\mathrm{m}$ and $9756\,\mathrm{m}$) with concentrations of cloud ice, cloud snow, and supercooled liquids that are $2-6$ times higher compared to the other two thunderstorm environments in these domains, as well as large precipitation amounts, strong large-scale vertical velocities, and increased shear. Wind-field$_{lowMF}$ thunderstorms (light blue pluses) have in general very low mass-field values, as seen in the lowest panel (domain K). All three bluish wind-field thunderstorm environments have low CAPE values across all domains with means of $194\,\mathrm{J\,kg^{-1}}$ in wind-field$_{CP}$ thunderstorms, $91\,\mathrm{J\,kg^{-1}}$ in wind-field$_{lowMF}$ thunderstorms, and $56\,\mathrm{J\,kg^{-1}}$ in the remaining wind-field environment. The common feature of these thunderstorms is increased horizontal wind speeds - hence the label 'wind-field´ thunderstorm environment.

The reddish mass-field thunderstorm environment has in all domains high mass-field variables with large CAPE values, high water vapor concentrations, and elevated $-10\,^\circ$C isotherm heights compared to other thunderstorms in the same domain. Differences between the two reddish lines occur almost exclusively in the surface-exchange variables. High surface-exchange values are characteristic of mass-field$_{SX}$ thunderstorms, which are characterized by high downward solar radiation ($> 320\,\mathrm{W\,m^{-2}}$), large latent heat fluxes ($> 140\mathrm{W\,m^{-2}}$), and upward oriented sensible heat fluxes. The thunderstorm sub-environments wind-field$_{lowMF}$ and mass-field$_{SX}$ both occur in conditions where the (sea) surface is hot relative to the air at $1000\,\mathrm{m}$ altitude, with an average temperature difference of $10.5\,\mathrm{K}$, while in mass-field thunderstorms (without sub-environment) the air mass at $1000\,\mathrm{m}$ is only about $3.9\,\mathrm{K}$ colder than the surface. Regarding the topographical influences, mass-field thunderstorms occur more often over land (higher land-sea mask values), and wind-field thunderstorms more often over the sea.

In each domain, at least one wind-field related thunderstorm environment and one mass-field related thunderstorm environment are found. The two major thunderstorm environments clearly separate from one another. Varying the number of clusters $k$ robustly finds similar results. With $k = 2$ only the two major thunderstorm environments are found. With $k > 3$ more and more clusters are found referring to an already existing thunderstorm environment revealing no additional meteorological insights.

**Table 2.** Cluster means for each thunderstorm environment and domain.

| | CAPE | CIN presence | Pressure (msl) | Water vapor (total column) | Temperature dew point (2 m) | -10 C isotherm height | Temp. diff. sfc (sst or skt) - 1000 m a.g.l. | Boundary layer height | Solar radiation (sfc, net, down) | Sensible heat flux (sfc, up) | Latent heat flux (sfc, up) | Wind speed at 10 m | Wind speed at cloud base | Cloud shear | Boundary layer dissipation | Vertical velocity (maximum, up) | Vertical liquids flux -8 C -12 C | Ice crystal mass -10 C -40 C | Snow mass -10 C -40 C | Supercooled liquids (total column) | Convective precipitation | Large-scale precipitation | Cloud size | Land-sea mask (0 = sea, 1 = land) | Elevation |
|---|---|---|---|---|---|---|---|---|---|---|---|---|---|---|---|---|---|---|---|---|---|---|---|---|---|
| Unit | $J\,kg^{-1}$ | binary | hPa | $kg\,m^{-2}$ | °C | m a.g.l. | K | m | $W\,m^{-2}$ | $W\,m^{-2}$ | $W\,m^{-2}$ | $m\,s^{-1}$ | $m\,s^{-1}$ | $m\,s^{-1}$ | $W\,m^{-2}$ | $Pa\,s^{-1}$ | $g\,Pa\,s^{-1}$ | $g\,m^{-2}$ | $g\,m^{-2}$ | $g\,m^{-2}$ | mm | mm | m | binary | m |
| **Coastal region** | | | | | | | | | | | | | | | | | | | | | | | | | |
| A mass-field | 297 | 0.59 | 1009.9 | 28.3 | 13.5 | 4533 | 3.5 | 440 | 164 | 4 | 50 | 5.1 | 9.7 | 13.3 | 2 | 0.587 | 3 | 56 | 80 | 29 | 0.049 | 0.005 | 7258 | 0.43 | 9 |
| A wind-field$_{lowMF}$ | 66 | 0.15 | 999.9 | 11.8 | 5.2 | 2582 | 9.3 | 1147 | 76 | 22 | 106 | 10.1 | 15.6 | 11.4 | 9 | 0.544 | 2 | 31 | 32 | 28 | 0.037 | 0.003 | 5700 | 0.34 | 5 |
| A wind-field$_{CP}$ | 158 | 0.38 | 1001.6 | 23.8 | 9.8 | 3815 | 5.2 | 966 | 69 | -8 | 71 | 9.8 | 17.4 | 27.9 | 18 | 1.42 | 36 | 214 | 335 | 116 | 0.109 | 0.062 | 8885 | 0.42 | 7 |
| B mass-field$_{SX}$ | 362 | 0.69 | 1009.8 | 24.5 | 12.9 | 4185 | 9.1 | 925 | 328 | 54 | 149 | 4.3 | 8.6 | 11.7 | 3 | 0.615 | 4 | 45 | 59 | 37 | 0.066 | 0.004 | 7331 | 0.88 | 43 |
| B mass-field | 305 | 0.58 | 1009.8 | 30.7 | 13.9 | 4699 | 2 | 331 | 65 | -16 | 22 | 4.5 | 10.2 | 15.1 | 2 | 0.674 | 7 | 96 | 153 | 36 | 0.057 | 0.014 | 7786 | 0.62 | 27 |
| B wind-field | 59 | 0.11 | 999.3 | 11.5 | 4.2 | 2463 | 7.9 | 1232 | 40 | -10 | 83 | 10.2 | 17.8 | 15.6 | 16 | 0.678 | 8 | 53 | 66 | 49 | 0.037 | 0.013 | 6036 | 0.55 | 15 |
| C mass-field$_{SX}$ | 454 | 0.77 | 1010.9 | 26.5 | 13.7 | 4441 | 9.6 | 994 | 341 | 59 | 166 | 3.8 | 7.9 | 10.5 | 3 | 0.586 | 3 | 39 | 57 | 35 | 0.065 | 0.003 | 7707 | 0.95 | 117 |
| C mass-field | 315 | 0.61 | 1010.4 | 30.7 | 14.3 | 4735 | 2.6 | 324 | 46 | -15 | 25 | 4 | 9.3 | 13.3 | 1 | 0.585 | 4 | 71 | 113 | 35 | 0.053 | 0.012 | 7407 | 0.71 | 79 |
| C wind-field | 43 | 0.07 | 1000.4 | 11.3 | 2.9 | 2317 | 6.8 | 1244 | 41 | -33 | 60 | 8.9 | 18.2 | 17.7 | 20 | 0.731 | 9 | 61 | 72 | 54 | 0.034 | 0.018 | 5806 | 0.85 | 81 |
| D mass-field | 410 | 0.63 | 1011.4 | 31.9 | 15.8 | 4902 | 4.1 | 383 | 138 | 10 | 57 | 3.7 | 8.7 | 14.5 | 1 | 0.731 | 3 | 59 | 89 | 25 | 0.053 | 0.005 | 7432 | 0.56 | 68 |
| D wind-field$_{lowMF}$ | 76 | 0.21 | 1004.2 | 13.9 | 7.4 | 2962 | 10.8 | 1204 | 103 | 46 | 156 | 10.2 | 15 | 14.6 | 9 | 0.771 | 3 | 48 | 50 | 29 | 0.064 | 0.004 | 6106 | 0.27 | 25 |
| D wind-field$_{CP}$ | 115 | 0.29 | 1007.1 | 18.2 | 8.2 | 3148 | 7.5 | 999 | 60 | -4 | 89 | 7.3 | 13.1 | 26.5 | 17 | 1.755 | 43 | 144 | 223 | 121 | 0.096 | 0.058 | 8059 | 0.77 | 155 |
| **Continental region** | | | | | | | | | | | | | | | | | | | | | | | | | |
| E mass-field$_{SX}$ | 329 | 0.57 | 1013.3 | 21.7 | 11.4 | 4071 | 13.2 | 1268 | 412 | 125 | 146 | 2.7 | 5.5 | 13.7 | 3 | 0.796 | 4 | 38 | 48 | 47 | 0.055 | 0.005 | 7725 | 0.99 | 769 |
| E mass-field | 424 | 0.61 | 1013.9 | 26.8 | 14.1 | 4471 | 5 | 343 | 59 | 2 | 35 | 2.8 | 7.2 | 15.9 | 2 | 0.661 | 3 | 53 | 80 | 28 | 0.044 | 0.007 | 7373 | 0.78 | 484 |
| E wind-field$_{CP}$ | 123 | 0.37 | 1008.1 | 17.1 | 8.2 | 3140 | 8.3 | 993 | 72 | 17 | 89 | 7.6 | 12.5 | 24.1 | 14 | 1.428 | 29 | 106 | 161 | 85 | 0.102 | 0.047 | 7974 | 0.63 | 308 |
| F mass-field$_{SX}$ | 408 | 0.68 | 1011.7 | 24.5 | 13.1 | 4261 | 10.5 | 1020 | 364 | 75 | 171 | 2.9 | 6.6 | 13 | 3 | 0.654 | 4 | 39 | 54 | 37 | 0.057 | 0.004 | 7574 | 0.99 | 310 |
| F mass-field | 380 | 0.66 | 1011 | 30.1 | 14.5 | 4656 | 3.3 | 317 | 33 | -13 | 27 | 2.8 | 8.9 | 17.4 | 3 | 0.922 | 11 | 104 | 175 | 42 | 0.073 | 0.023 | 8406 | 0.96 | 276 |
| F wind-field | 58 | 0.16 | 1001.9 | 13.6 | 5.8 | 2763 | 7.3 | 1153 | 75 | -11 | 78 | 7.9 | 15.7 | 20.7 | 17 | 0.926 | 17 | 68 | 95 | 66 | 0.054 | 0.026 | 6579 | 0.83 | 178 |
| G mass-field$_{SX}$ | 451 | 0.73 | 1012.7 | 25.1 | 13 | 4318 | 10.5 | 1015 | 372 | 73 | 176 | 2.7 | 6 | 11.2 | 2 | 0.598 | 4 | 39 | 57 | 37 | 0.06 | 0.004 | 8007 | 1 | 406 |
| G mass-field | 318 | 0.59 | 1012 | 29.2 | 13.9 | 4612 | 3.2 | 269 | 29 | -14 | 24 | 2.4 | 8.5 | 14.4 | 2 | 0.702 | 7 | 91 | 156 | 34 | 0.056 | 0.012 | 7992 | 1 | 383 |
| G wind-field | 64 | 0.18 | 1003.6 | 11.6 | 2.6 | 2285 | 6.2 | 1272 | 49 | -38 | 50 | 7.3 | 16.3 | 24.8 | 27 | 0.921 | 19 | 83 | 113 | 68 | 0.046 | 0.033 | 6071 | 1 | 359 |
| **Alpine-central region** | | | | | | | | | | | | | | | | | | | | | | | | | |
| H mass-field$_{SX}$ | 486 | 0.7 | 1013.5 | 21.4 | 12 | 3932 | 9.9 | 832 | 342 | 63 | 171 | 1.7 | 3.9 | 15.4 | 3 | 0.773 | 5 | 49 | 77 | 40 | 0.074 | 0.009 | 8778 | 1 | 918 |
| H mass-field | 287 | 0.47 | 1013.1 | 21 | 10.2 | 3736 | 3.5 | 147 | 21 | -4 | 13 | 1.4 | 4.3 | 18.1 | 3 | 0.727 | 4 | 53 | 80 | 39 | 0.046 | 0.013 | 7611 | 1 | 855 |
| H wind-field$_{CP}$ | 122 | 0.37 | 1007.3 | 14.7 | 6.8 | 2687 | 4.1 | 471 | 24 | -11 | 23 | 2.8 | 7.8 | 35.6 | 31 | 2.608 | 164 | 182 | 397 | 239 | 0.088 | 0.15 | 8085 | 1 | 910 |
| I mass-field$_{SX}$ | 481 | 0.73 | 1012.5 | 25.6 | 13.8 | 4405 | 10.8 | 1036 | 392 | 79 | 188 | 2.2 | 5.1 | 11.7 | 2 | 0.647 | 4 | 37 | 56 | 37 | 0.061 | 0.004 | 7964 | 1 | 437 |
| I mass-field | 397 | 0.65 | 1011.5 | 28.9 | 14.2 | 4592 | 3.2 | 225 | 28 | -10 | 25 | 2.1 | 7.5 | 14.7 | 2 | 0.717 | 5 | 69 | 119 | 31 | 0.057 | 0.008 | 7950 | 1 | 314 |
| I wind-field$_{CP}$ | 139 | 0.4 | 1005.6 | 15.8 | 6 | 2902 | 4.7 | 554 | 30 | -6 | 20 | 3.2 | 6.4 | 32.6 | 24 | 1.811 | 75 | 143 | 274 | 149 | 0.101 | 0.096 | 8731 | 1 | 676 |
| **Mediterranean region** | | | | | | | | | | | | | | | | | | | | | | | | | |
| J mass-field | 759 | 0.77 | 1012.9 | 31.4 | 17.3 | 4919 | 6.8 | 457 | 173 | 17 | 74 | 4.3 | 7.8 | 14.6 | 1 | 0.607 | 2 | 42 | 64 | 22 | 0.035 | 0.005 | 7213 | 0.25 | 66 |
| J wind-field$_{lowMF}$ | 97 | 0.26 | 1008.7 | 16 | 8.7 | 3260 | 11.3 | 973 | 121 | 43 | 144 | 7.8 | 10.2 | 13.5 | 3 | 0.601 | 2 | 37 | 40 | 29 | 0.042 | 0.004 | 5905 | 0.14 | 38 |
| J wind-field$_{CP}$ | 308 | 0.66 | 1008.2 | 26.7 | 13.2 | 4132 | 8.3 | 763 | 68 | 20 | 109 | 8.3 | 12.4 | 24.9 | 6 | 1.654 | 35 | 169 | 295 | 93 | 0.149 | 0.07 | 9907 | 0.23 | 59 |
| K mass-field | 930 | 0.84 | 1012.7 | 31 | 17.6 | 4916 | 7.8 | 501 | 218 | 31 | 94 | 3.6 | 6.4 | 12.2 | 1 | 0.598 | 2 | 38 | 57 | 24 | 0.04 | 0.004 | 7328 | 0.4 | 142 |
| K wind-field$_{lowMF}$ | 103 | 0.27 | 1008.2 | 15.7 | 8.5 | 3227 | 11 | 877 | 128 | 39 | 127 | 6.5 | 8.9 | 13.1 | 3 | 0.62 | 2 | 35 | 37 | 31 | 0.042 | 0.004 | 5992 | 0.24 | 82 |
| K wind-field$_{CP}$ | 316 | 0.67 | 1006.5 | 25.3 | 12.9 | 4034 | 7.8 | 716 | 65 | 14 | 92 | 7.2 | 11.4 | 25.1 | 8 | 1.636 | 28 | 160 | 248 | 83 | 0.155 | 0.057 | 9756 | 0.39 | 129 |
| L mass-field | 933 | 0.9 | 1011 | 31.2 | 17.3 | 4859 | 7.5 | 448 | 189 | 23 | 90 | 3.5 | 6.2 | 14.6 | 1 | 0.691 | 3 | 52 | 81 | 29 | 0.058 | 0.006 | 8218 | 0.47 | 110 |
| L wind-field$_{lowMF}$ | 114 | 0.27 | 1008.5 | 16.2 | 8.3 | 3212 | 10.3 | 697 | 107 | 32 | 102 | 5.1 | 7.3 | 17 | 2 | 0.763 | 3 | 50 | 56 | 32 | 0.046 | 0.007 | 6645 | 0.32 | 96 |
| L wind-field$_{CP}$ | 268 | 0.61 | 1005.4 | 23.8 | 11.4 | 3819 | 7 | 668 | 45 | 4 | 75 | 6.2 | 10.9 | 28.3 | 11 | 2.052 | 59 | 174 | 284 | 124 | 0.162 | 0.082 | 9720 | 0.61 | 205 |

CAPE: convective available potential energy, CIN: convective inhibition, msl: mean sea level, a.g.l.: above ground level,

Temp. diff. sfc (sst or skt) - 1000 m a.g.l.: Temperature difference between the surface and 1000 m a.g.l., where sfc is either sea surface temperature or skin temperature.

In summary, there are two major thunderstorm environments in Europe (wind-field thunderstorms and mass-field thunder-storms) and three sub-environments thereof. Thunderstorm environments are found by applying cluster analysis on 12 domains. A decision tree is developed to differentiate the thunderstorm environments using their driving meteorological categories.

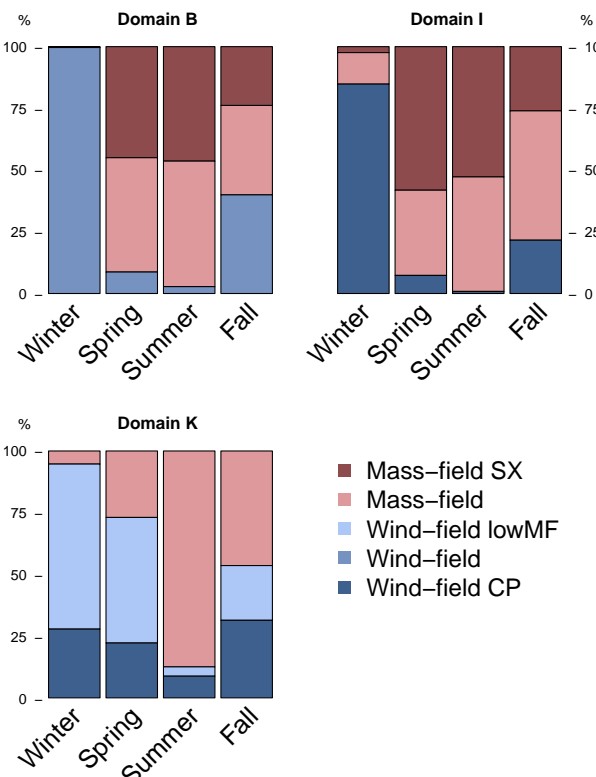

**Figure 6.** Seasonal variation of the thunderstorm environments within three representative domains (B, I, K; panels). Bars are equally high because the same number of observations from each season is used. Results for each domain are based on cluster analysis estimated on each domain separately using local scaling values.

## 4.3 Seasonal differences between thunderstorm environments in Europe

The stacked barplots in Fig. 6 show how many lightning observations from each season belong to a given thunderstorm environment. As the data set is built to have the same number of observations from each season, the bars are equally high. 275 The absolute numbers of observations per domain are given in Table 1. In all domains, winter (DJF) is dominated by wind-field thunderstorm environments (blues) and summer (JJA) by mass-field thunderstorm environments (reds). Spring and fall are transitional seasons with varying proportions. If a domain has two wind-field thunderstorm environments (e.g., domain K), there is often a dominant thunderstorm environment with a more pronounced annual cycle (wind-field$_{lowMF}$) and a smaller environment (wind-field$_{CP}$) with less seasonality.

The map in Fig. 7 spatially compares barplots that are estimated individually on each domain using local mean and standard deviations for scaling. The polygon colors indicate which domains are similar to one another (Sect. 4.1) and hence more comparable as they are scaled with similar values (baselines, Table 1). In every domain, wind-field thunderstorms (blues) dominate in winter (first bar) and contribute to a varying fraction of thunderstorms in spring and fall, which is higher the more maritime a domain gets (domains A, D, J, K, L). Mass-field thunderstorms (reds) always dominate in summer (third bar) and over the mainland also in spring and fall. The presented cluster analysis with $k = 3$ has in all maritime domains (A, D, J, K, L) two wind-field thunderstorm environments present, and in all domains at the mainland (B, C, E, F, G, H, I) two mass-field thunderstorms. This reveals the importance of wind-field thunderstorms over the seas. Higher $k$ results in further splitting of the displayed thunderstorm environments to have at least two wind-field thunderstorm environments and two mass-field thunderstorm environments present in every domain. In Fig. 7, wind-field thunderstorms in the southern domains (E, H, I) are accompanied by enhanced cloud-physics variables (wind-field$_{CP}$) which is remarkable in the alpine-central region (H, I) as all thunderstorms have in general very high cloud-physics variables there. As the presented thunderstorm characteristics are *relative* to other thunderstorms in each domain they are not directly comparable because some of them refer to very different baselines (Sect. 4.1). The mediterranean region (J, K, L) for example refers to much higher overall mass-field variables (Fig. 3) than the coastal domains (e.g., domain A).

In summary, wind-field thunderstorms dominate the cold season and are more important over the sea while mass-field thunderstorms dominate the warm season and are more important over the mainland.

## 4.4 Comparability of the thunderstorm environments

The thunderstorm environments are identified *relative* to the general meteorological conditions during lightning in each domain, and the question remains how similar the thunderstorm environments of the same name from different domains are.

To make the thunderstorm environments more comparable, a principal component analysis is estimated on all cluster means from every domain using the same scaling (Fig. 8). Again, the first two principal components are displayed on the axes explaining together about 70 % of the variance (PC 3 explains additionally 13.4 %) and the labeled arrows (loadings) indicate the contribution of each variable to the variance in the respective direction. Each domain (letters) is represented by three colored circles, the thunderstorm environments found there. First of all, the figure shows that the two major thunderstorm environments, wind-field thunderstorms and mass-field thunderstorms, clearly separate as the bluish and reddish circles are located in different parts of the figure. The difference between the mass-field thunderstorms is small as the reddish circles gather close to one another. Their major difference is in the surface-exchange variables that separate the light red mass-field environment from the dark red mass-field$_{SX}$ sub-environment, which becomes more relevant in PC 3. Wind-field thunderstorms are more diverse as the bluish circles spread widely. They dominate the cold season, where climatologically fewer thunderstorms occur. Hence, the scarce cold-season thunderstorms in Europe originate in very diverse thunderstorm conditions while frequent summertime mass-field thunderstorms originate in similar weather patterns.

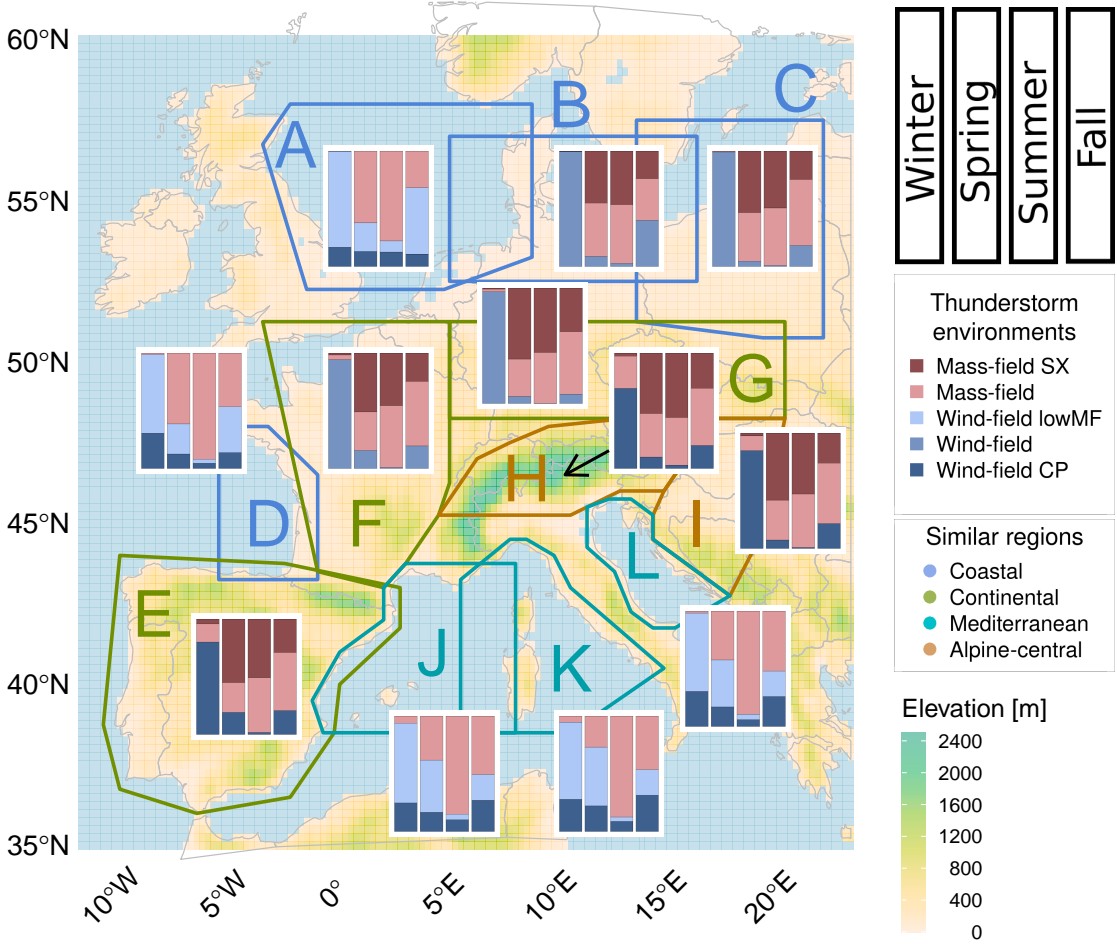

**Figure 7.** Spatio-temporal analysis of thunderstorm environments in Europe. Barplots are based on cluster analyses that are individually estimated on each domain using local scaling values (as in Fig. 6). The colors of the domain borders indicate similarity between some domains.

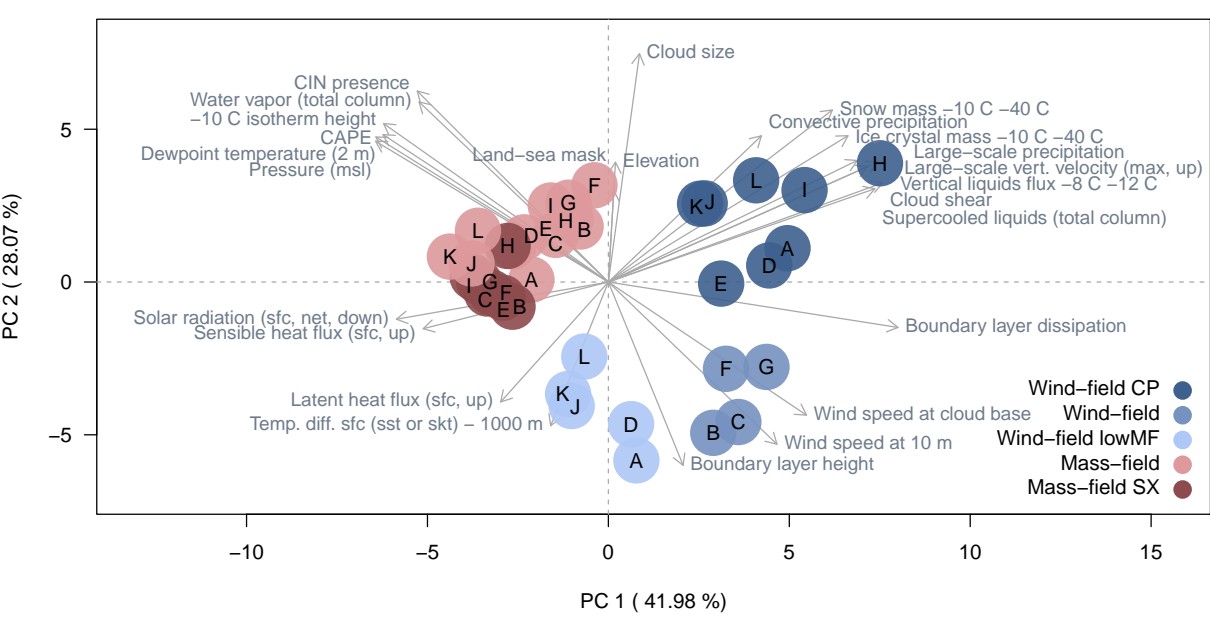

**Figure 8.** Comparison between all clusters (i.e., thunderstorm environments) from all domains using the same scaling. The figure shows a biplot based on a PCA estimated on Table 2 and taking the first two principal components (PC) as axes. Each point represents a cluster and is colored and labeled according to its thunderstorm environment and domain. The labeled arrows (loadings) indicate the contribution of each considered ERA5 variable to the variance in the respective direction.

## 5 Discussion

Regional lightning differences are described by four distinct regions: coastal, continental, mediterranean, and alpine-central. Thunderstorm characteristics in different meteorological conditions are provided by the thunderstorm environments (wind-field thunderstorms and mass-field thunderstorms plus sub-environments).

Other authors have also investigated thunderstorm conditions. The variables important for our wind-field thunderstorms are similar to Mäkelä et al. (2013)'s investigations of winter lightning in Finland. Their investigations revealed the importance of vertical temperature difference between the surface and mid-troposphere (700/500 hPa) and low-tropospheric wind shear, while CAPE was no useful predictor. Another classification was performed by Fujii et al. (2013) in Japan, who found that the number of winter lightning strokes and the probability of high-current lightning strokes, group into the *storm type* and *inactive type* dependent on the $-10\,°C$ isotherm height. Market et al. (2002) show in their thundersnow climatology over the contiguous United States that lightning associated with snowfall originates in seven different meteorological settings. This supports our finding of the very diverse wind-field thunderstorm conditions. Sherburn and Parker (2014) coined the term HSLC-thunderstorms, which are meteorological environments of high-shear ($\geq 18\,\mathrm{m\,s}^{-1}$ at $0-6$ km) and low-CAPE ($\leq 500\,\mathrm{J\,kg}^{-1}$) capable to produce lightning in all seasons and at all times of day in the United States. Considering the overall lower CAPE values in Europe, this HSLC concept relates to our wind-field thunderstorm environments, especially the sub-environment with particularly low mass-field values (wind-field$_{\mathrm{lowMF}}$).

High shear, as it is characteristic of wind-field thunderstorms, results in tilted clouds. Thus, charge separation occurs along a slanted path within the cloud, and the charge centers are also separated horizontally. This is known as the *tilted charge hypothesis* (Takeuti et al., 1978; Brook et al., 1982; Engholm et al., 1990; Williams, 2018; Takahashi et al., 2019; Wang et al., 2021), and is often described for cold season thunderstorms. We think that this tilt provides sufficiently large distances between the charge centers to cause lightning, even though the clouds are shallow and CAPE is low, with cluser-means of approximately $100\,\mathrm{J\,kg}^{-1}$ or less (Table 2). In mass-field thunderstorms, on the other hand, CAPE often exceeds $300\,\mathrm{J\,kg}^{-1}$ (factor 3 and more), and horizontal wind speeds are lower, so that charge can separate along a more upright path. Both types require conditionally unstable parts in the temperature and moisture profiles, and thus have increased mass-field values compared to conditions without lightning (Morgenstern et al., 2022). Thunderstorms with tilted clouds are referred to as *wind-field* thunderstorms to emphasize that lightning is unlikely to occur when, in addition to low CAPE values, the horizontal wind speeds are also low.

Stucke et al. (2022) relate our two major thunderstorm environments as described in Morgenstern et al. (2022) to upward lightning at two alpine towers and find that most upward lightning occurs in wind-field thunderstorm conditions. Thus, wind-field thunderstorms pose a particular risk to tall infrastructure and should be considered when determining the lightning threat to wind farms. If Stucke et al. (2022)'s relation between wind-field thunderstorms and upward lightning (i.e., lightning to tall structures) holds also for flat terrain and for the thunderstorm sub-environment lowMF, then offshore wind farms are at particular risk from wind-field lightning. The lightning protection standard IEC 61400-24 (2019) could be improved by additionally including the proportion of wind-field thunderstorms at sites considered for wind farms. Currently, only the local

lightning density, the height of the structure, and an environmental factor (i.e., factors for winter lightning, terrain slope, and elevation) are taken into account (IEC 61400-24, 2019). The concept of thunderstorm environments introduced here is superior to the idea of *winter lightning* versus *summer lightning* because it takes regional and seasonal differences into account. The concept is easily transferable to many locations because it is independent of static thresholds as they are for example used by March et al. (2016), Montanyà et al. (2016), or Sherburn and Parker (2014).

In general, thunderstorm frequencies under different synoptic conditions are often described (e.g., Wapler and James, 2015; Enno et al., 2014; Bielec, 2001; Kolendowicz, 2006) and regional thunderstorm differences are often subject of classical climatologies as mentioned in the introduction. For the Baltic countries, Enno et al. (2013) found three distinct thunderstorm regions (continental, transitional, maritime) similar to some of our thunderstorm regions (continental, coastal). The Baltic countries probably have similar proportions of wind-field thunderstorms and mass-field thunderstorms as domain C because this domain covers parts of Lithuania and Latvia. However, Enno et al. (2013)'s thunderstorm regions indicate that the transition between maritime and continental thunderstorms is just a few dozen kilometers inland. Hence, a higher proportion of wind-field related thunderstorm environments compared to domain C is expected in the Baltic countries, as most parts of these countries are close to the coast.

For the UK and Ireland, Hayward et al. (2022) conduct a regional cluster analysis aiming to identify areas where the seasonal distributions of lightning densities differ. This climatology nicely complements our approach with the PCA (Sect. 4.1) because it has a better resolution and covers adjacent regions, where EUCLID data does not fulfill our quality requirements. The regions Hayward et al. (2022) distinguished, are continental, coastal, or marine. Large parts of the UK and Ireland are classified as marine regions and produce thunderstorms in winter. Hence, the UK and Ireland are probably similar to our domain A with many wind-field$_{lowMF}$ thunderstorms. One characteristic of thunderstorms in this environment are large temperature differences between the surface and the overlying air mass, which coincides with Hayward et al. (2022)'s description of lightning in marine and coastal regions.

This study is limited by the resolution of the data used. Finer distinctions in the thunderstorm sub-environments are expected with higher resolution in the reanalysis data. More details in the model topography might lead to a more precise thunderstorm differentiation in complex terrain and a convection-resolving resolution could reveal more details about the meteorological characteristics of the thunderstorms themselves, not just the environments in which they occur. But this requires reanalysis on a scale finer than currently available. Further, a longer time series in the EUCLID data would allow more regions to be analyzed. Scandinavia has been excluded from this investigation because the scarcity of lightning in winter caused the sample size over the investigated 11 years to be too small for an unbiased statistical analysis. These limitations affect only the thunderstorm sub-environments; the main results (wind-field thunderstorms and mass-field thunderstorms) are expected to remain the same.

Based on the results presented, several new research questions arise that are beyond the scope of this paper. How often are wind turbines or other tall structures struck by which thunderstorm environment? What is the relationship between thunderstorm environments and lightning properties such as the lightning duration, transferred charge, polarity, or channel length? Is the decision tree (Fig. 4) valid for other extratropical regions? Are there other thunderstorm (sub-) environments in other climate zones such as the tropics? It would also be interesting to model lightning probability maps for each thunderstorm

environment in each season, and to investigate the proportion of all thunderstorms in a given region that occur in a particular thunderstorm environment.

## 6 Conclusions

This study investigates seasonal and regional differences of meteorological environments in which lightning occur in Europe.
Highly destructive lightning damages often occur in seasons and regions where lightning is climatologically unlikely. They pose a challenge for lightning risk assessments because time series of lightning observations are often short and the meteorological conditions for lightning in the cold season are not well understood. This study explicitly includes infrequent lightning conditions by considering an equal number of lightning observations from each season. EUCLID lightning data are combined with meteorological ERA5 data to answer two research questions: "Are there regions in Europe, where thunderstorms occur
under similar meteorological conditions?" and "What characterizes thunderstorms in different meteorological environments and how do they vary seasonally across Europe?" Using coarse but consistent reanalysis data, this study paves the way for lightning reconstructions by providing tools to diagnose favorable lightning conditions.

Using principal component analysis, the European territory can be divided into four regions where the atmospheric conditions for thunderstorms are similar throughout the year: The alpine-central region with thick clouds, large cloud particle
concentrations, and strong large-scale vertical velocities relative to the other regions; the mediterranean region with increased mass-field variables; the coastal region with increased wind speeds; and the continental region with in general average conditions and increased solar radiation relative to the other regions. Cluster analysis is performed individually on 12 domains in Europe to find and describe different thunderstorm environments and to name them according to their characteristics compared to other thunderstorms in that domain. A decision tree is developed to easily distinguish the thunderstorm environments
(Fig. 4).

There are two major thunderstorm environments – wind-field thunderstorms and mass-field thunderstorms – and three sub-environments thereof. Mass-field thunderstorms are characterized by increased CAPE values, the presence of CIN, large $2\,\mathrm{m}$ dew point temperatures, high $-10\,^{\circ}\mathrm{C}$ isotherm heights, and high mean sea level pressure relative to other thunderstorms in the same domain. The release of CAPE results in a quasi-vertical separation of the charged particles. Mass-field thunderstorm
environments occur mostly in the warmer seasons and always in similar weather conditions and are more important over the European mainland. The mass-field$_{\mathrm{SX}}$ sub-environment is associated with enhanced surface-exchange (SX) variables such as solar radiation and sensible heat flux and accounts for about half of the mass-field thunderstorms on the European mainland.

The other major thunderstorm environment, wind-field thunderstorms, originate in more diverse weather conditions, but share the characteristics of average or low values in mass-field variables and elevated or average values in wind-field variables
(high wind speeds at different heights, strong large-scale vertical velocities, large cloud shear, and increased boundary layer dissipation) relative to other thunderstorms in the same domain. In this environment, CAPE is a poor predictor of whether lightning will occur because it is generally small. High wind speeds and shear cause the charged particles to be separated along slanted paths. Wind-field thunderstorms dominate the cold season, especially winter, and are more important over the

sea. Sometimes the cloud-physics (CP) variables are additionally enhanced leading to the wind-field$_{CP}$ thunderstorm sub-environment with large cloud sizes, increased concentrations of cloud particles (snow, ice, supercooled liquids), and large amounts of precipitation. Another sub-environment, wind-field$_{lowMF}$, is characterized by particularly low mass-field variables (low MF) and often occurs over the sea.

The process-oriented view of different thunderstorm environments challenges the traditional idea of *winter lightning* versus *summer lightning* and makes it easier to compare similar processes with different magnitudes in different regions. In summary, this study shows that lightning in Europe originate in different meteorological environments, that winter lightning is not just a rarer sibling of summer lightning, and provides a decision tree to easily differentiate thunderstorm environments in Europe independent of a seasonal criterion or static thresholds.

**Appendix A:  Additional details to Sect. 4.1 "Regional differences between thunderstorms in Europe"**

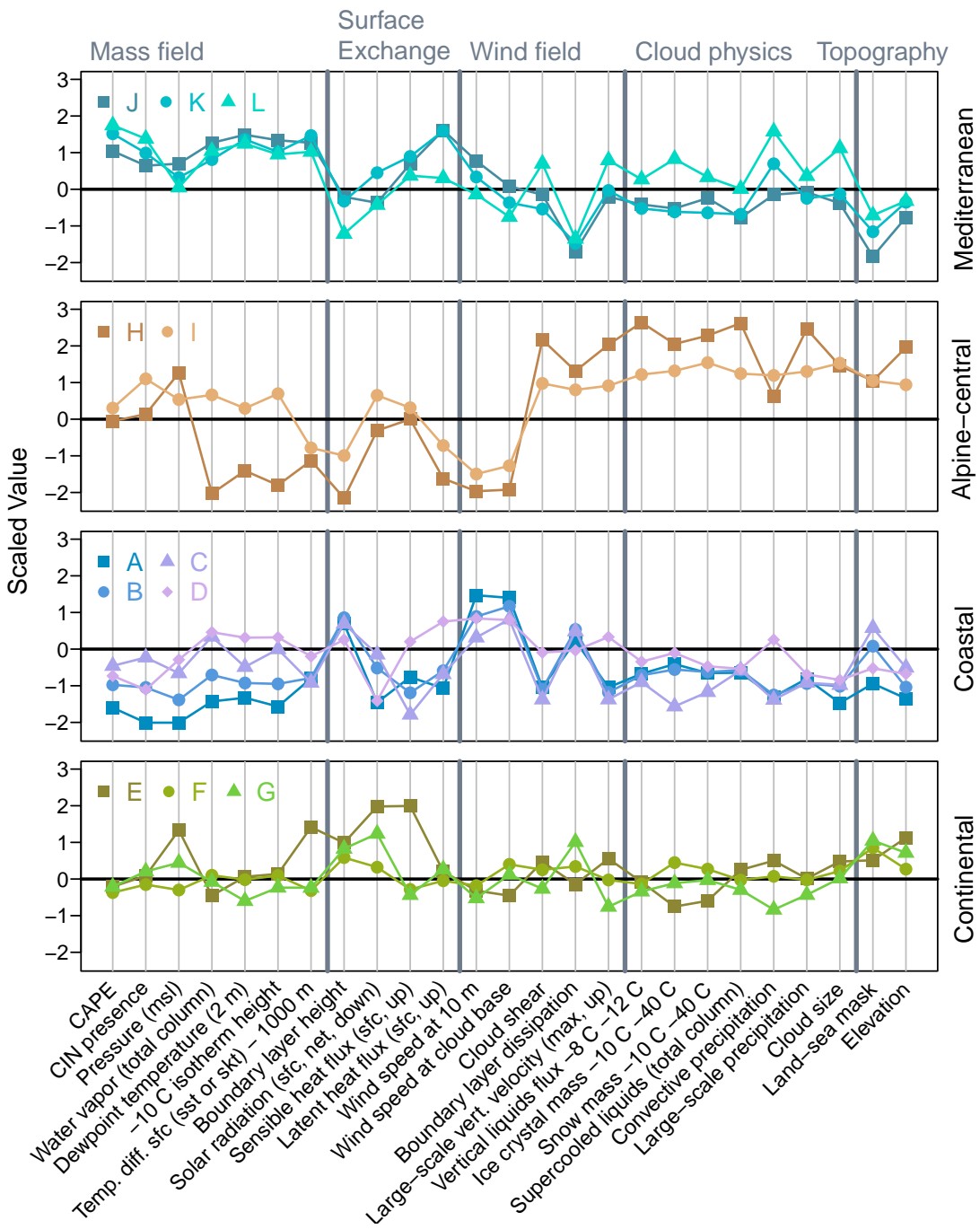

**Figure A1.** Additional details to the parallel coordinate plot in Fig. 3 based on Table 1. The panels are the basis for the means in Fig. 3.

## Appendix B:  Additional details to Sect. 4.2 "Thunderstorm environments"

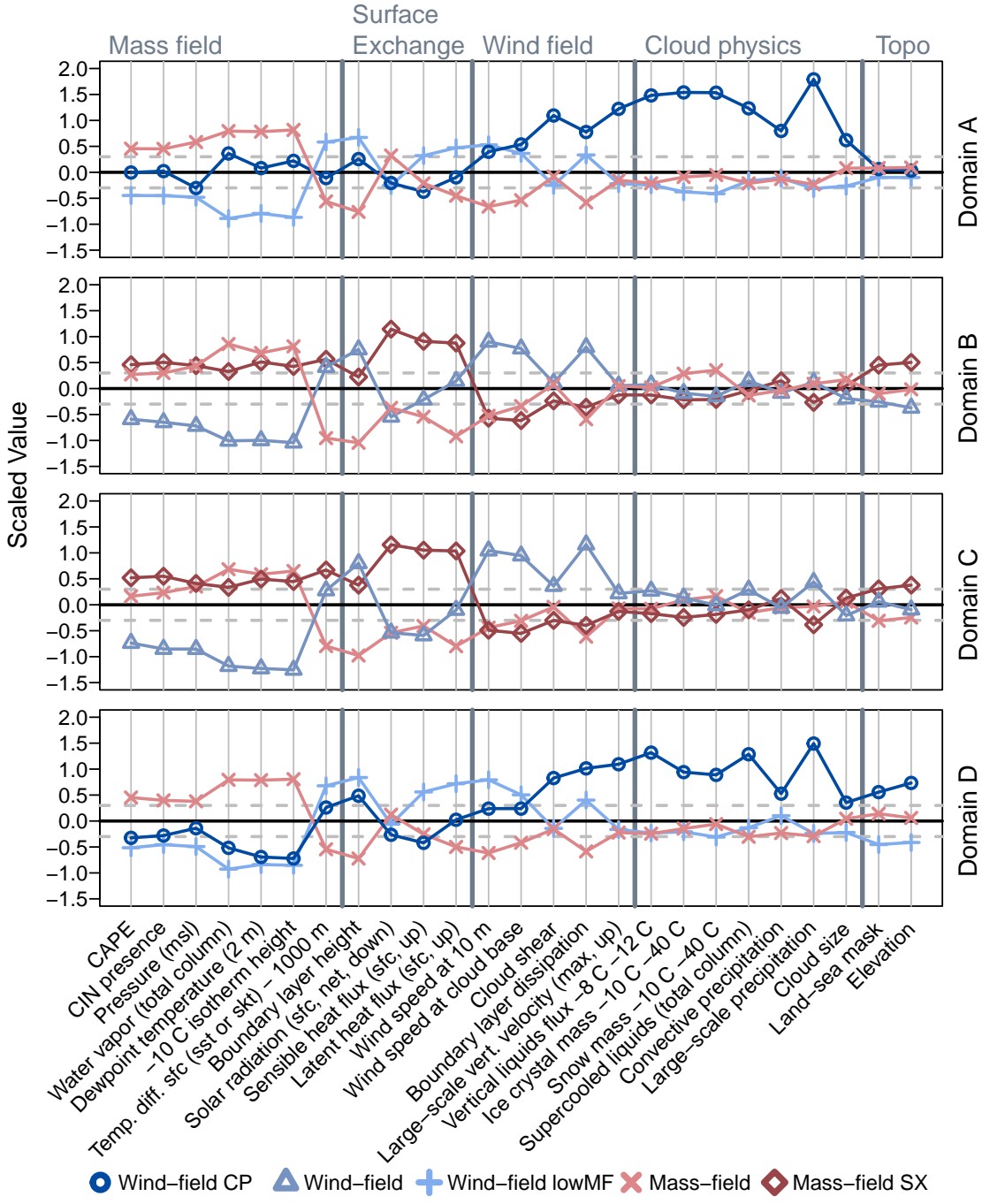

**Figure B1.** Additional details to Fig. 5. Here: Cluster means for the coastal domains. Numbers are given in Table 2.

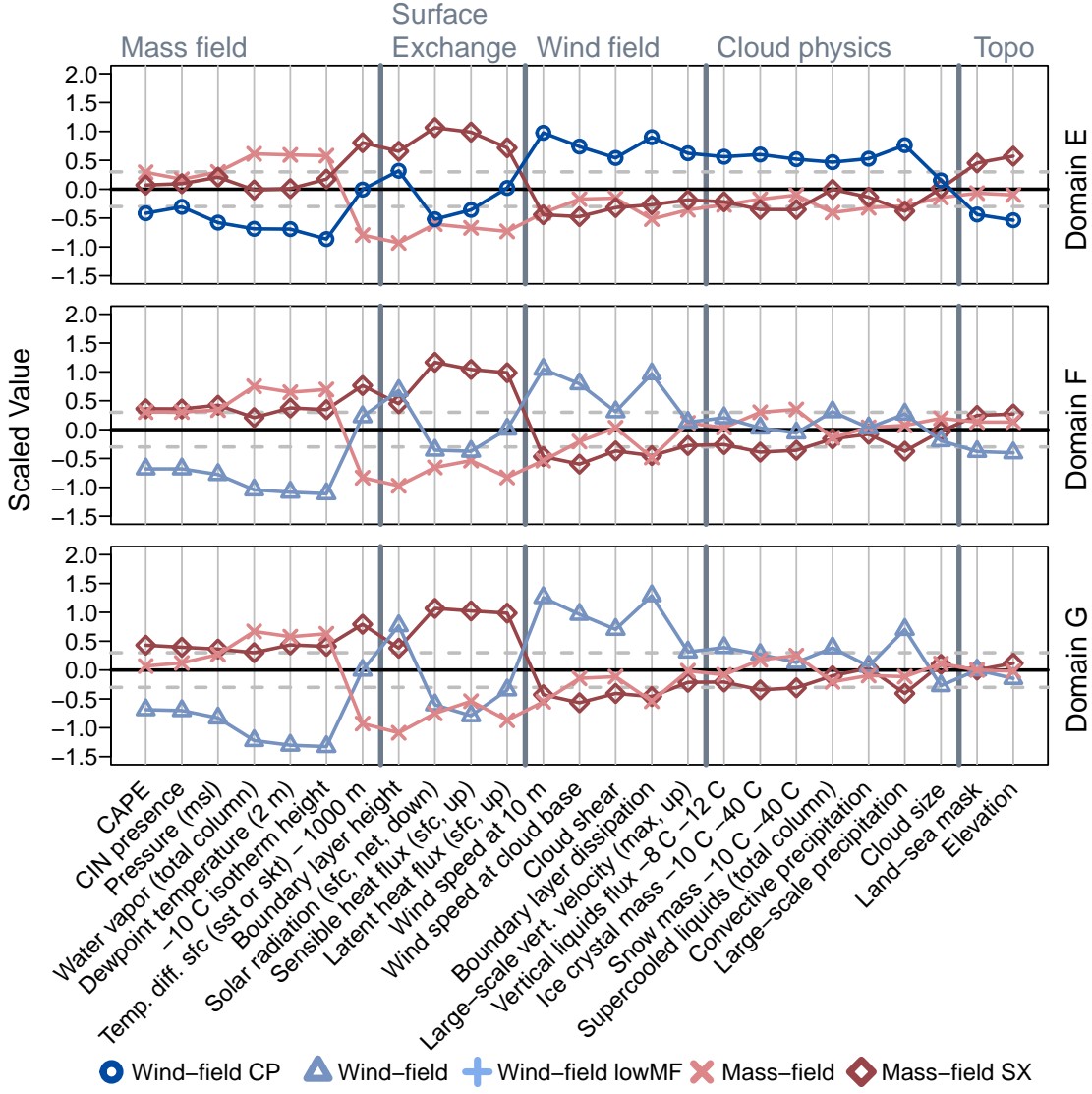

**Figure B2.** Additional details to Fig. 5. Here: Cluster means for the continental domains. Numbers are given in Table 2.

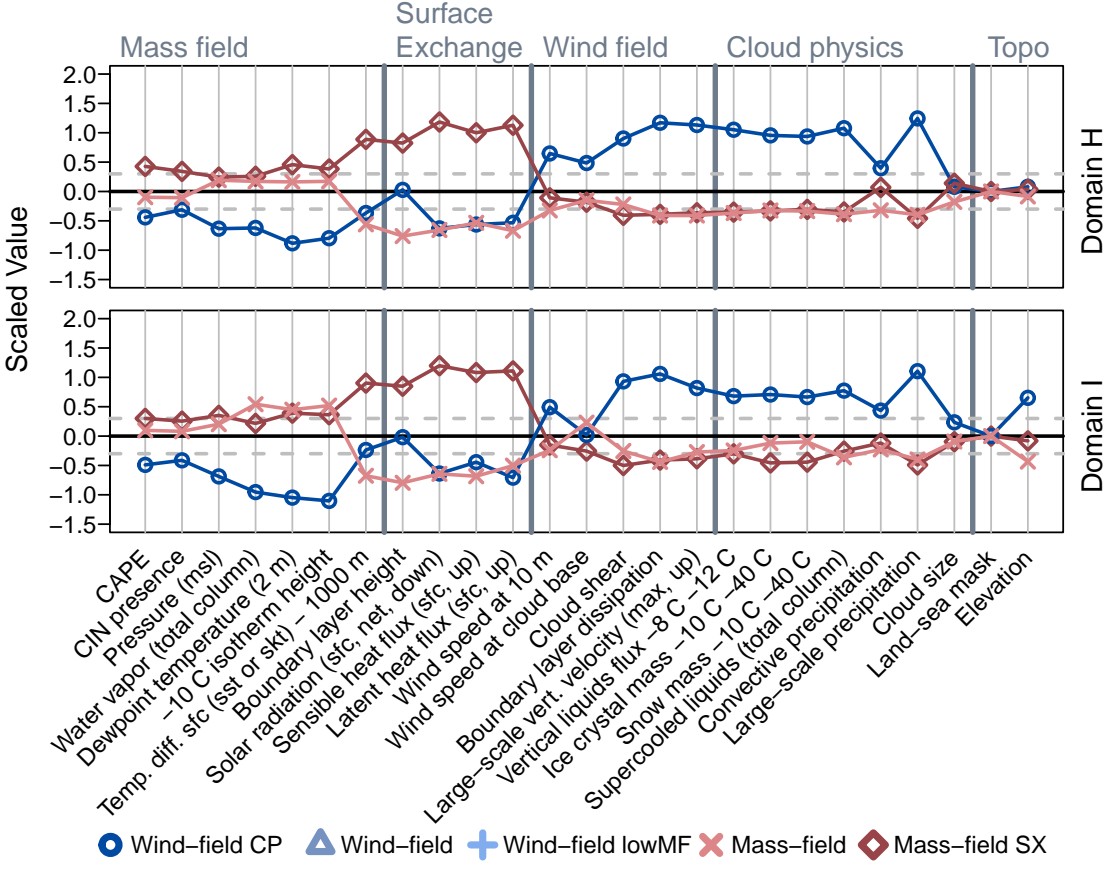

**Figure B3.** Additional details to Fig. 5. Here: Cluster means for the alpine-central domains. Numbers are given in Table 2.

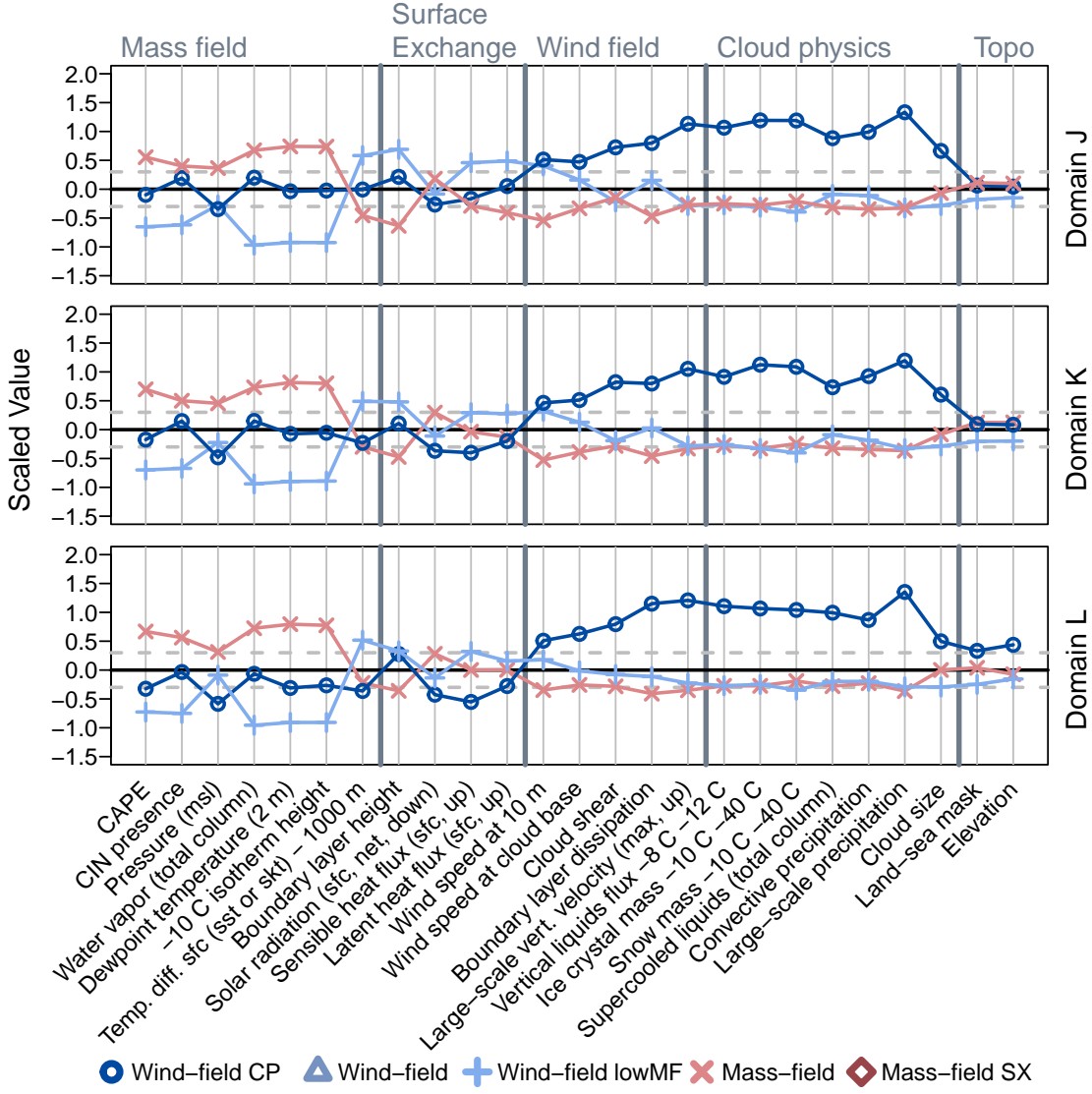

**Figure B4.** Additional details to Fig. 5. Here: Cluster means for the mediterranean domains. Numbers are given in Table 2.

*Code and data availability.* The supplementary material (Morgenstern et al., 2023) contains a R script to reproduce the core findings and main figures along with the required data for the presented sample, the domain definitions, a precise variable description, and the two tables. ERA5 data are freely available from the Copernicus Climate Change Service (C3S) Climate Data Store (Hersbach et al., 2020; https://cds.climate.copernicus.eu, last access: 30 September 2022). This study uses ERA5 hourly data 1959 to present on single level and model level (https://doi.org/10.24381/cds.adbb2d47, Hersbach et al., 2018; https://confluence.ecmwf.int/display/CKB/How+to+download+ERA5, last access: 25 November 2022). The results contain modified Copernicus Climate Change Service information for 2010–2020. Neither the European Commission nor ECMWF is responsible for any use that may be made of the Copernicus information or data it contains. EUCLID (Poelman et al., 2016; Schulz et al., 2016) data are available on request from ALDIS (Austrian Lightning Detection & Information System, aldis@ove.at) or Siemens BLIDS (Blitzinformationsdienst, fees may apply).

Calculations are performed using R (https://www.R-project.org/, R Core Team, 2021), Python 3 (https://www.python.org/, Van Rossum and Drake, 2009), and CDO (Climate Data Operator; https://doi.org/10.5281/zenodo.3539275, Schulzweida, 2019). Specifically, the following packages are used: ncdf4 (https://CRAN.R-project.org/package=ncdf4, Pierce, 2019), sf (simple features; https://r-spatial.github.io/sf/index.html, Pebesma, 2018), stars (https://CRAN.R-project.org/package=stars, Pebesma, 2020), rnaturalearth (https://CRAN.R-project.org/package=rnaturalearth, South, 2017), data.table (https://github.com/Rdatatable/data.table/wiki, Dowle and Srinivasan, 2020), colorspace (https://colorspace.r-forge.r-project.org/, Stauffer et al., 2009), and xarray (Hoyer and Hamman, 2017). The netCDF4 data format is used (https://doi.org/10.5065/D6H70CW6, Unidata, 2020).

*Author contributions.* DM performed the investigation, wrote the software, visualized the results, and wrote the paper. IS, TS, and DM performed the data curation, built the data set, and derived variables based on ERA5 data. TS contributed coding concepts. GJM provided support for the meteorological analysis, data organization, and funding acquisition. AZ supervised the formal analysis and interpretation of the statistical methods. AZ, GJM, and TS are the project administrators and supervisors. All authors contributed to the conceptualization of this paper, discussed the methodology, evaluated the results, and commented on the paper.

*Competing interests.* The authors declare no competing interests.

*Acknowledgements.* We are grateful to Gerhard Diendorfer, Wolfgang Schulz, and Hannes Pichler from ALDIS for data support and discussions about lightning physics and to EUCLID for providing the LLS data.

This research has been supported by the Österreichische Forschungsförderungsgesellschaft (FFG) with grant no. 872656 and the Austrian Science Fund (Fonds zur Förderung der wissenschaftlichen Forschung, FWF) with the grants no. P 31836 and no. P 35780-NBL and with financial support from the State Tirol, Austria.

The computational results presented have been achieved in part using the Vienna Scientific Cluster (VSC).

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
