# Peer review of "Thunderstorm Environments in Europe"

_EGUsphere, 2022_

## Author Comment (AC1)

Johannes Dahl
Co-Editor
–
Referee 1 and 2

**Date**
March 3, 2023

**Revision of WCD-2022-1453**

Dear Johannes Dahl, dear referees,

please find attached the revised version of our manuscript *"Thunderstorm Types in Europe"* (WCD-2022-1453).

Thank you very much for the constructive and helpful feedback from you, the associate editor, and the referees. Based on this feedback, the manuscript has been revised. We feel that this substantially improved our contribution. The most important changes are:

- *Renaming*.
  The former concept of 'thunderstorm types´ is renamed to 'thunderstorm environments´.

- *Motivation*.
  Our motivation to diagnose thunderstorm environments based on larger-scale reanalysis data is stated more clearly.

- *Figures*.
  Figures are updated for better readability.

- *Discussion and limitations*.
  The discussion is extended to better show possible applications of our research, to extrapolate our results to other regions, and a more in-depth discussion of the limitations is provided.

All changes and additions are explained in much more detail in the point-to-point reply on the next pages.

Best regards,

The authors

**Referee 1**

*The overall quality of this paper is good. The research uses objective statistical methods (PCA and K-means clustering) to simplify atmospheric variables, thunderstorm types and spatially similar regions to good effect. This enhances the understanding of thunderstorm types present in different regions of Europe and explores their seasonal variations. In particular, this method allows the identification of less frequently occurring thunderstorm types which pose a risk to particular types of infrastructures. The thunderstorm types identified by this method had not yet been defined in this way across a wide and variable geographic area, so this identification, therefore, complements and enhances the current understanding of thunderstorm behaviour in Europe. The methodology to identify thunderstorm types may now be used at regional resolution and in other areas to continue to add to knowledge.*

Thank you.

*1) The Abstract – consider explaining more clearly the rationale for undertaking this research. The strikes to tall infrastructure are mentioned in the first sentence as the main reason for undertaking the research. Tall infrastructure is then not mentioned very often in the main body of the paper. It is suggested that either the authors include more discussion of the relevance of their results to tall infrastructure so it is more prominent in the mind of the reader. Alternatively, the abstract could be reworded so that the emphasis is taken off tall infrastructure and it could be mentioned merely as an example of the potential benefits / end users of this research.*

ACTION: Tall infrastructure is no longer mentioned in the introduction. A new sentence in the motivation leads the reader to the potential benefits of our research for the protection of tall infrastructure (same issue as referee 2, number 6).

> Line 1 (Abstract): »Meteorological environments favorable for thunderstorms are studied across Europe, including rare thunderstorm conditions from seasons with climatologically few thunderstorms.«

> Line 15: »Thunderstorms during the cold season are generally rare but pose a serious threat to wind turbines and other tall structures because it has been observed that lightning strikes to tall infrastructure have no or only a weak annual cycle whereas lightning in general has a pronounced annual cycle (Stucke et al., 2022; Matsui et al., 2020).«

*2) The discussion makes good points and relates this research back to the literature well. However, some greater depth would enhance this part of the paper nicely. Examples of points which could be expanded:*

ACTION: Large parts of the discussion are enhanced by more details. The differences are best tracked in the diff-file.

*a) The identification of the thunderstorm types could be related back to the results of previous studies with similar domains (coastal, continental) and seasonal variations in lightning occurrence – the Baltic and the UK are mentioned in particular and perhaps the authors could extrapolate from their results which thunderstorm type and by extension atmospheric conditions, might produce a risk from lightning in the regions / seasons identified in these countries (where there might be wind farms located). This should then fit in nicely with the author's statement that these already identified regions compliment their research by bringing together all this information in one place.*

ACTION 1: More details on the consequences of our results to lightning threat at wind farms is provided.

> Line 327 : »Thus, wind-field thunderstorms pose a particular risk to tall infrastructure and should be considered when determining the lightning threat to wind farms. If Stucke et al. (2022)'s relation between wind-field thunderstorms and upward lightning (i.e., lightning to tall structures) holds also for flat terrain and for the thunderstorm sub-environment noMF, then offshore wind farms are at particular risk from wind-field lightning. The lightning protection standard IEC 61400-24 (2019) could be improved by additionally including the proportion of wind-field thunderstorms at sites considered for wind farms. Currently, only the local lightning density, the height of the structure, and an environmental factor (i.e., factors for winter lightning, terrain slope angle, and elevation) are taken into account (IEC 61400-24, 2019). The concept of thunderstorm environments introduced here is superior to the idea of winter lightning versus summer lightning as it takes into account regional and seasonal differences. The concept is easily transferable to many locations as it is independent of static thresholds as they are used for example by March et al. (2016), Montanyà et al. (2016), or Sherburn and Parker (2014).«

ACTION 2: Additional paragraphs relate our research better to similar studies in the Baltic countries, the UK, and Ireland. The new parts are:

> Line 342: »The Baltic countries probably have similar proportions of wind-field thunderstorms and mass-field thunderstorms as domain C because this domain covers parts of Lithuania and Latvia. However, Enno et al. (2013)'s thunderstorm regions indicate that the transition between maritime and continental thunderstorms is just a few dozen kilometers inland. Hence, a higher proportion of wind-field related thunderstorm environments compared to domain C is expected in the Baltic countries, as most parts of these countries are close to the coast.«

> Line 350: »The regions Hayward et al. (2022) distinguished, are continental, coastal, or marine. Large parts of the UK and Ireland are classified as marine regions and produce thunderstorms in winter. Hence, the UK and Ireland are probably similar to our domain A with many wind-field noMF thunderstorms. One characteristic of thunderstorms in this environment are large temperature differences between the surface and the overlying air mass, which coincides with Hayward et al. (2022)'s description of lightning in marine and coastal regions.«

**b) The authors mention that this research is limited by the resolution of atmospheric data. Do the authors have any suggestions to tackle this for further research? Are there any potential higher-resolution features (mountain ranges) within the identified domains / regions that may influence the thunderstorm types as a result?**

ACTION 1: The possible improvements and expected results from increased resolution are discussed in more detail (same issue as in referee 2 number 2 and 4).

> Line 356 : »This study is limited by the resolution of the data used. Finer distinctions in the thunderstorm sub-environments are expected with higher resolution in the reanalysis data. More details in the model topography might lead to a more precise thunderstorm differentiation in complex terrain and a convection-resolving resolution could reveal more details about the meteorological characteristics of the thunderstorms themselves, not just the environments in which they occur. But this requires reanalysis on a scale finer than currently available. Further, a longer time series in the EUCLID data would allow more regions to be analyzed. Scandinavia has been excluded from this investigation because the scarcity of lightning in winter caused the sample size over the investigated 11 years to be too small for an unbiased statistical analysis. These limitations affect only the thunderstorm sub-environments; the main results (wind-field thunderstorms and mass-field thunderstorms) are expected to remain the same.«

ACTION 2: Further research questions are formulated in the discussion.

> Line 367: »Is the decision tree (Fig. 4) valid for other extratropical regions? Are there other thunderstorm (sub-) environments in other climate zones such as the tropics? It would also be interesting to model lightning probability maps for each thunderstorm environment in each season, and to investigate the proportion of all thunderstorms in a given region that occur in a particular thunderstorm environment.«

**3) The text of the paper does contain some grammatical errors which are noted where identified by the reviewer. However, there may be others which have not been identified, so the authors should check again the text for grammar using relevant grammar assistance technology if necessary. The authors may also wish to check whether it is the convention in their field/this journal to use American-English or English-English as there are some instances of "summarize" instead of "summarise" or "fall" instead of "autumn" which the journal may or may not have a preference for.**

Using appropriate words and omitting grammatical errors increases the readability of the paper and better conveys the content. We are grateful for the many suggestions from you and from the copy editors later in the process. Weather and climate dynamics accepts all standard varieties of English but strives for consistency within each article. We decided to apply american english and to use the Oxford serial comma.

`https://www.weather-climate-dynamics.net/submission.html#english`

*4) L11 – Perhaps provide examples of meteorological settings.*

The many meteorological settings in which thunderstorms in Europe may originate are discussed in much detail later in the introduction. The first paragraph aims to provide a motivation and to lead the reader into the topic. Details and references are kept to a minimum in this introductory paragraph to better provide the big picture.

NO ACTION.

*5) L12 – Suggest replacing "Plenty" with "Numerous". If "Plenty" is to remain suggest amendment to "Plenty of".*

ACTION 1: Words are replaced as suggested.
ACTION 2: Further changes to that sentence: 'most of them' is replaced by 'many' and 'types and rare lightning' is omitted.

>    Line 13: »Numerous lightning climatologies are available but many focus on the dominant characteristics and seasons while infrequent thunderstorm conditions are often neglected.«

*6 a) L34 – Suggest replacing "at land and advected to the sea" with "over land and advected out to sea" or "onshore and advected out to sea".*

*6 b) L35 – Perhaps instead of "it endures longer" which suggests a singular long-lasting lightning flash, replace with "where lightning activity endures longer".*

*6 c) L35 – Suggest replace "water surface is" with "sea surface temperatures are".*

ACTION: The words are replaced as suggested.

>    Line 37: »Nighttime offshore lightning [...] is explained by convection initiated over land and advected out to sea, where lightning activity endures longer as the sea surface temperatures are unaffected by nighttime cooling.«

*7) L46 – Suggest replace "to use the" with "the use of".*

ACTION: The words are replaced as suggested.

>    Line 47: »Mallick et al. (2022) even suggests the use of sea surface temperature as proxy for seasonal lightning forecasts.«

*8) L56 – Do you mean to ask the question of how thunderstorm characteristics vary by meteorological conditions across Europe or how thunderstorm occurrence varies by meteorological characteristics across Europe... Please make the wording of this question a little clearer.*

ACTION: The wording of this question is rephrased for clarity (same issue as in referee 2 number 2).

>    Line 71: »Are there regions in Europe, where thunderstorms occur under similar meteorological conditions?«

*9) L61 – suggest change "second one (Sect. 3)" to "second (Sect. 3)" The word "one" is unnecessary and removal may improve the flow of the sentence.*

ACTION: The word is omitted.

>    Line 74: »Applying similar methods as in Morgenstern et al. (2022a), principal component analysis finds the answer to the first question, and *k*-means clustering to the second (Sect. **??**).«

*10) L64 – Suggest change "The found thunderstorm types" to "These thunderstorm types".*

ACTION 1: The words are replaced as suggested.

ACTION 2: Other occurrences of the phrase are changed as well.

> Line 78: »These thunderstorm environments are then analyzed seasonally (Sect. 4.3) and compared to one another (Sect. 4.4).«

> Line 266: »A decision tree is developed to differentiate the thunderstorm environments using their driving meteorological categories.«

> Line 292 (subsection title): »Comparability of the thunderstorm environments«

> Figure 8 (caption): »Comparison between all clusters (i.e., thunderstorm environments) from all domains using the same scaling.«

*11) L87/88 – Suggest review of sentence grammar / rewording here to make clearer. The use of the word "As" to start the sentence does not seem quite right.*

ACTION: The paragraph is rephrased.

> Line 108: »Lightning data are provided by the European Cooperation for Lightning Detection(EUCLID, Schulz et al., 2016, Poelman et al., 2016), a cooperation of several local lightning location systems (LLS) in Europe. Only cloud-to-ground lightning flashes between 2010–2020 are considered, as this period is most stable regarding the hardware and software configuration of the network. If at least one lightning flash occurred within an ERA5 cell in a given hour, the whole cell-hour is regarded as one lightning observation.«

*12) L100 – "France and Belgium, being a less homogeneous but very representative domain." Representative of what? Please consider making this clearer for the reader.*

'Representative' relates to Europe in general, because Europe consists of a variety of different landscapes, such as flat terrain, hills, and coastlines. As this part is causing confusion it is omitted.

ACTION: Omitting confusing parts.

> Line 120: »Domain F covers large parts of France and Belgium, being a topographically less homogeneous domain.«

*13) L123 – "In the following, only one sample is discussed, as all repetitions led to qualitatively the same results." Please consider providing greater detail on how the repetitions were compared and found to be the same.*

The sample size in each domain consists of at least 5320 observations, i.e. at least 1330 observations per season. These large sample sizes are expected to be representative. To account for the unlikely case, that the drawn samples are not representative, the sampling procedure is repeated 50 times. This leads to 50 sets of results (including figures) for each domain. Each set of results is analyzed as described in the paper. The figures are visually compared (especially Figures 6, 5, and 2). If at all, only minor differences occur between most samples. Hence, the samples are representative.

ACTION: Greater detail on the comparison between the samples is given.

> Line 142: »For robustness, the whole analysis is performed on 50 different samples in each domain. A visual comparison of the resulting figures reveals qualitatively the same results between these repetitions. Hence, the samples are representative and it is sufficient to discuss only one sample in the following.«

**14) L157 – consider changing "are considered as 'baselines' there." To "are considered as 'baselines' " The word "there" does not seem to be necessary.**

ACTION: The word is omitted as suggested.

> Line 178: »These are typical values for thunderstorms throughout the year for the respective domains and are considered as 'baselines'.«

**15) L160 – "Meteorological similar domains gather close to one another" perhaps this should read something like "Domains with similar meteorological characteristics are represented in close proximity to one another within this diagram"**

ACTION: The sentence is rephrased similarly to what is suggested.

> Line 183: »Domains with similar meteorological characteristics are located in close proximity to one another within this diagram.«

**16) Figure 3 – This is a nice way to represent these results, however it is a little busy which makes interpretation more difficult. If possible, some improvements should be considered to simplify the plot for the benefit of the reader. Suggest experimenting with; removing or making smaller the point symbols, removing the domain letter labels to the right of the chart which can extend the x-axis (these can be moved to figure caption instead), increasing the size of the y-axis.**

Symbols are included to help readers with color deficiency.

ACTION 1: The reading experience for people with color deficiency is improved by stating that symbols and colors refer to the same features.

> Figure 2 (caption): »Domains with similar characteristics are indicated with the same color and symbol and labeled accordingly (legend).«

> Figure 3 (caption): »Similar domains are summarized into regions having the same color and symbol.«

ACTION 2: Line annotations are removed, the legend is expanded and moved to the bottom, category annotation is moved outside of plotting region, darker, and no longer italic.

[Figure]

Figure 1: Left: Previous figure. Right: Updated figure (Figure 3).

**17) L188 – Suggest change from "With this the spatial different thunderstorm conditions in Europe" to "These spatially different thunderstorm conditions in Europe"**

This short paragraph aims at summarizing the main findings to help quick readers to get the essence without going too much into detail. Therefore, the structure of the sentence is kept and only the grammar is corrected.

ACTION: Corrected grammar.

> Line 211: »With this, the spatially different thunderstorm conditions in Europe are described and summarized into four regions, …«

**18) Figure 4 is really good, shows the results very clearly for the reader to then understand the main text discussion. Suggest making the CP, noMF labels a bit more prominent so the distinction between the types is immediately obvious to the reader.**

ACTION: The index size is increased.

[Figure]

Figure 2: Left: Previous figure. Right: Updated figure (Figure 4).

**19) L208 – Style wise it is not usual to include we, or I in a research paper and the tone is a little informal. Suggest changing "Now we dive deeper into the characteristics of the found thunderstorm types" to something like "More detailed analysis of the thunderstorm type characteristics is undertaken".**

ACTION 1: The words are replaced as suggested.

ACTION 2: Further occurrences of 'we' or 'our' are rephrased.

> Line 103: »To ease interpretation, physical-based categories group the variables:«

> Line 136: »For each lightning cell-hour, ERA5 data at the respective cell and from the last full hour is taken to capture the build-up of the thunderstorms.«

> Line 158: »In this study, the first two PC's are used as axes for a so-called biplot to visualize the variance in the 25-dimensional data.«

> Line 163: »The optimal number of clusters $k$ for the data used in this study is derived from the sum of the squared residuals and ranges from 2 to 4. The results for $k = 3$ are presented in detail, and the results for $k$ 2 and $\geq$ 4 are also described.«

> Line 172: »Then thunderstorm environments are found by $k$-means clustering and a decision tree is presented to differentiate them (Sect. 4.2).«

> Line 175: »… and whether some of the 12 domains can be grouped …«

> Line 231: »A more detailed analysis of the thunderstorm environments is undertaken by investigating …«

> Line 290: »In summary, wind-field thunderstorms dominate the cold season and are more important over the sea while …«

*20) Figure 5 – Similar to figure 3 in places it is difficult to interpret due to being quite busy. Wind field noMF in particular has a rather faint colour which becomes obscured by others. Please consider again removing or reducing the size of the point symbols or changing the colour scheme to make the graph clearer to read.*

ACTION: The colors are more intense, the symbol size is decreased, the category annotations are darker,the y-axis labels and the secondary x-axis labels are moved out, and the line width is increased.

[Figure]

Figure 3: Left: Previous figure. Right: Updated figure (Figure 5).

*21) 241 – "Now that the thunderstorm types are found" tone is a bit informal. Suggest rewording this sentence.*

ACTION 1: The sentence is omitted.

ACTION 2: Another occurrence of that phrase is reworded.

> Line 231: »A more detailed analysis of the thunderstorm environments is undertaken by …«

> Line 364: »Based on the results presented, several new research questions arise …«

*22) L246 – Where it states "there is often a bigger one" consider replacing the word "one" with something clearer or rewording the sentence, perhaps "there is often a dominant type"*

ACTION: The words are replaced.

> Line 272: »If a domain has two wind-field thunderstorm environments (e.g., domain K), there is often a dominant thunderstorm environment with a more pronounced annual cycle (wind-field$_{noMF}$) and a smaller type (wind-field$_{CP}$) with less seasonality.«

**23) Figure 6 – include y-axis labels or y-axis explainer in the figure caption. To make it clear whether this refers to absolute counts, is a percentage or scaled like earlier plots.**

ACTION: The y-axes are changed to display percentages and labeled accordingly.

[Figure]

Figure 4: Left: Previous figure. Right: Updated figure (Figure 6).

**24) Figure 7 – J, K and L domains with their barplots are a little cramped, suggest investigating offsetting the barplots to edge of the figure and indicating their domain using arrows to see if this improves things a little.**

ACTION: The barplots are moved and an arrow is included. The legend label is changed to 'Thunderstorm environments'.

[Figure]

Figure 5: Left: Previous figure. Right: Updated figure (Figure 7).

**25) L277 – Consider replacing "little thunderstorms" with "fewer thunderstorms"**

ACTION: The words are replaced as suggested.

>Line 304: »They dominate the cold season, where climatologically fewer thunderstorms occur.«

**26) L300 – It is not ideal to start a sentence with the word "And". Suggest rewording this.**

ACTION 1: The sentence is rephrased.

ACTION 2: Other sentences starting with "and" are also rephrased.

>Line 4 (abstract): »occurring mainly in winter; and mass-field thunderstorms, characterized by«

>Line 309: »Thunderstorm characteristics in different meteorological conditions are provided by«

>Line 348: »For the UK and Ireland, Hayward et.al. (2022) conduct a regional cluster analysis.«

**27) L313 – Consider changing "lightning is climatological unlikely" to "lightning is climatologically unlikely"**

ACTION: The words are replaced as suggested.

>Line 373: »Highly destructive lightning damages often occur in seasons and regions where lightning is climatologically unlikely.«

**28) L317 – Please remove full stop from: Europe?".**

ACTION: The full stop is removed.

>Line 378: »"What characterizes thunderstorms in different meteorological environments and how do they vary seasonally across Europe?"«

**29) L336 – Consider changing "origins" to "originates"**

ACTION 1: The word is replaced as suggested.

ACTION 2: Other occurrences of 'origins' are replaced.

>Line 315: »Market et al. (2002) show in their thundersnow climatology over the contiguous United States that lightning associated with snowfall originates in seven different meteorological environments. This supports our finding of the very diverse wind-field thunderstorm conditions.«

>Line 407: »In summary, this study shows that lightning in Europe originate in different meteorological environments, …«

***30) Please consider the comments made regarding the figures in the main text of the paper with respect to the figures in the appendix.***

ACTION 1: Appendix 1 has varying colors, the symbol size is decreased, the y-axis labels are moved, and the category annotations are darker.
ACTION 2: Appendix 2 - 5, same changes as to Fig. 6

[Figure]

Figure 6: Left: Previous figure. Right: Updated figure (Appendix A 1).

[Figure]

Figure 7: Left: Previous figure. Right: Updated figure.
Only Appendix B 1 is included in this answer letter. Analog changes are applied to Appendix B 2-4.

**Referee 2**

*1) According to the title and the abstract, the present work "Thunderstorm Types in Europe" by Deborah Morgenstern, Isabell Stucke, Georg J. Mayr, Achim Zeileis, and Thorsten Simon shall provide a possibility to distinguish thunderstorms independent of the season and fixed thresholds (like CAPE). The work is mainly motivated by the fact that thunderstorms also occur in winter and for high infrastructure even no significant seasonal correlation can be identified. For this purpose, model parameters of ERA5 reanalysis data and EUCLID lightning data are statistically analyzed and it has been possible to describe significant differences between at least two situations in which thunderstorms occur. The results suggest that other atmospheric processes are important for the development of winter thunderstorms compared to thunderstorms in summer. The results allow to compare thunderstorm types for different regions independent of the season.*

*2) The results presented are relevant on this point, showing that obviously different atmospheric processes enable thunderstorms. However, there are also important limitations. For example, the analyzed data are only coarsely resolved and accordingly do not allow to resolve the thunderstorms. At best, they reflect the large-scale environment and parameterization of thunderstorms.*

The paper aims to identify meteorological environments in which lightning occurs. Unfortunately, the title and abstract of the manuscript did not reflect that goal clearly enough and have been changed (more details at number 5).

The benefit of reanalysis data is their consistency and availability over several decades. Defining thunderstorm environments using larger-scale meteorology allows finding these environments also in coarser reanalysis data. This allows to calculate reconstructions in the future.

ACTION 1: The first research question is rephrased for clarity (same issue as in referee 1 number 8).

> Line 71: »Are there regions in Europe, where thunderstorms occur under similar meteorological conditions?«

ACTION 2: The aim to find thunderstorms in coarse reanalysis data is now explicitly stated in the data section.

> Line 81: »Two data sets are incorporated in this study (cf. Morgenstern et al., 2022a): meteorological reanalysis data (ERA5, Sect. 2.1) representing meteorological environments, and lightning observations (EUCLID, Sect. 2.2).«

> Line 89: »ERA5 provides consistent data on the state of the atmosphere at a scale larger than individual thunderstorms. It directly contains and allows to derive additional variables related to various atmospheric processes relevant to lightning: The presence of differently sized cloud particles and strong motions leading to collision and subsequent separation of the particles.«

ACTION 3: The discussion on the limitations is extended (same issue as in referee 1 number 2b).

> Line 357: »This study is limited by the resolution of the data used. Finer distinctions in the thunderstorm sub-environments are expected with higher resolution in the reanalysis data. More details in the model topography might lead to a more precise thunderstorm differentiation in complex terrain and a convection-resolving resolution could reveal more details about the meteorological characteristics of the thunderstorms themselves, not just the environments in which they occur. But this requires reanalysis on a scale finer than currently available.«

*3) Moreover, the parameter selection is also arbitrary (line 79 and following is not well-founded enough in this context) and accordingly not necessarily suitable to characterize thunderstorm situations.*

The variable set focuses on processes leading to the formation and separation of charge within a thundercloud. To allow statistical analysis, only variables that are not highly correlated to one another are included in the variable set. The variable selection was accompanied by an extended literature research and explorative data analysis. The variables we have considered more closely are provided in Table 1.

ACTION: The data section is extended to better motivate the variable set.

> Line 89: »ERA5 provides consistent data on the state of the atmosphere at a scale larger than individual thunderstorms. It directly contains and allows to derive additional variables related to various atmospheric processes relevant to lightning: The presence of differently sized cloud particles and strong motions leading to collision and subsequent separation of the particles. Almost one hundred such variables were computed and exploratively analyzed. By eliminating highly correlated variables that provide limited additional information, a set of 25 variables remains (Table 2). The set is indicative of substantial clouds (e.g., moisture, updrafts, precipitation, cloud size), charge transfer (e.g., ice, snow, supercooled liquids), and charge separation (e.g., shear, wind, CAPE, CIN). As some required variables are not directly available at ERA5 (https://www.doi.org/10.24381/cds.adbb2d47, accessed 2023-02-15), they are derived using also the model level data. These additionally derived variables include variables such as …«

Table 1: Meteorological variables directly available at ERA5 or derived from it (*). A subset of this extended variable set is used in the paper.

| Variable | Unit | Variable | Unit |
|---|---|---|---|
| Boundary layer dissipation | $J\,m^{-2}$ | Pressure (at model levels) * | hPa |
| Boundary layer height | m | Rain water (at model or pressure levels) | $kg\,kg^{-1}$ |
| CAPE | $J\,kg^{-1}$ | Rain water (total) | $kg\,m^{-2}$ |
| CAPE shear | $m^2\,s^{-2}$ | Rain water between -8 C -12 C, or -10 C -20 C, or -20 C -40 C * | $kg\,m^{-2}$ |
| CIN presence * | binary | Sea ice cover | (0 - 1) |
| Cloud base height | m a.g.l. | Sea surface temperature | K |
| Cloud cover (high, medium, low) | (0 - 1) | Shear (10m – cloud top, or 10m – cloud base, or within cloud) * | $m\,s^{-1}$ |
| Cloud size * | m | Shear direction (10m – cloud top, or 10m – cloud base, or within cloud) * | ° |
| Cloud top height * | m | Snow cover | (0 - 1) |
| Coastline distance * | km | Snow depth | mm |
| Convective prcp. 1h-sum (all or snow only) | mm | Solar radiation (sfc) | $J\,m^{-2}$ |
| Day of year * | numeric | Solid hydrometeors (at model or pressure levels) | $kg\,kg^{-1}$ |
| Dew point temperature 2m | K | Solid hydrometeors between -8 C -12 C, or -10 C -20 C, or -20 C -40 C * | $kg\,m^{-2}$ |
| Energy flux divergence | $W\,m^{-2}$ | Solid hydrometeors, total | $kg\,m^{-2}$ |
| Evaporation | mm | Solids between -8 C -12 C (ice + solid) * | $kg\,m^{-2}$ |
| Geopotential (at model or pressure levels) | $m^2\,s^{-2}$ | Supercooled liquids, total | $kg\,m^{-2}$ |
| Graupel (total) | $kg\,m^{-2}$ | Surface latent heat flux | $J\,m^{-2}$ |
| Hail | mm | Surface sensible heat flux | $J\,m^{-2}$ |
| Humidity (at model or pressure levels) | $kg\,kg^{-1}$ | Temperature (2m, skin, 500m, 750m, 1000m) | K |
| Humidity (total) | $kg\,m^{-2}$ | Temperature (at model or pressure levels) | K |
| Humidity between -10 C -20 C, or -20 C -40 C * | $kg\,m^{-2}$ | Temperature difference 500m and sst or skt * | K |
| Ice crystals (at model or pressure levels) | $kg\,kg^{-1}$ | Temperature difference 750m and sst or skt * | K |
| Ice crystals between -8 C -12 C, or -10 C -20 C, or -20 C -40 C * | $kg\,m^{-2}$ | Temperature difference 1000m and sst or skt * | K |
| Ice crystals divergence | $kg\,m^{-2}\,s^{-1}$ | Thermal radiation (sfc) | $J\,m^{-2}$ |
| Ice crystals, total | $kg\,m^{-2}$ | Topography (sfc geopotential height) * | m |
| Isotherm heights (0 C, -10 C, or -20 C) * | m a.g.l. | Vertical velocity (at model or pressure levels) | $Pa\,s^{-1}$ |
| Land-sea cover (binary or fraction) * | | Vertical velocity (maximum) * | $Pa\,s^{-1}$ |
| Large scale prcp. 1h-sum (all or snow only) | mm | Wind direction (10m, cloud base, cloud top, at model levels) * | ° |
| Liquid water (at model or pressure levels) | $kg\,kg^{-1}$ | Wind divergence | $s^{-1}$ |
| Liquid water (total) | $kg\,m^{-2}$ | Wind gust (10m) | $m\,s^{-1}$ |
| Liquid water between -8 C -12 C, or -10 C -20 C, or -20 C -40 C * | $kg\,m^{-2}$ | Wind speed (10m, cloud base, cloud top, at model levels) * | $m\,s^{-1}$ |
| Liquid water divergence | $kg\,m^{-2}\,s^{-1}$ | Wind u (1000m, 3000m, 5000m, cloud base, cloud top) * | $m\,s^{-1}$ |
| Liquids between -8 C -12 C (liquid + rain) * | $kg\,m^{-2}$ | Wind u (at model or pressure levels) | $m\,s^{-1}$ |
| Liquids updraft between -8 C -12 C (liquids + rain, or liquids only) * | $kg\,Pa\,s^{-1}$ | Wind u 10m | $m\,s^{-1}$ |
| Maximum precipitation rate (hour) | $kg\,m^{-2}\,s^{-1}$ | Wind v (at model or pressure levels) | $m\,s^{-1}$ |
| Moisture divergence | $kg\,m^{-2}\,s^{-1}$ | Wind v (1000m, 3000m, 5000m, cloud base, cloud top) * | $m\,s^{-1}$ |
| Orography (sfc geopotential) | $m^2\,s^{-2}$ | Wind v 10m | $m\,s^{-1}$ |
| Pressure (at mean sea level or at surface) | hPa | | |

CAPE = convective available potential energy, CIN = convective inhibition, prcp. = precipitation, sfc = surface, sst = sea surface temperature, skt = skin temperature.

**4) The results are accordingly only of limited use for further research, since, contrary to what is suggested in the abstract and summary, by no means only fixed thresholds or seasonal parameters are used for thunderstorm forecasting, especially not when using high-resolution weather models.**

Our research aims at diagnosing thunderstorms based on larger-scale meteorology to pave the way for lightning reconstructions based on coarse reanalysis data. We aim to describe meteorological environments favorable to produce thunderstorms. This is different from thunderstorm forecasting using high-resolution weather models where atmospheric processes are dynamically calculated on a fine grid. We aim at improving the meteorological background considered in the lightning protection standard IEC 61400-24 to provide better risk assessments in the future. The current standard uses static thresholds (March et al. 2016, IEC 61400-24 2019, Méndez et al. 2018) and applies the idea of "winter lightning" versus "summer lightning". Our research shows that this seasonal representation is incomplete and suggests using thunderstorm environments instead.

ACTION 1: The relevance of our research for future lightning protection is stated more clearly by including the limitations of the current lightning protection standard in the introduction and the discussion (similar issue as referee 1, number 2).

> Line 56: »The 2018 update of the lightning protection standard for wind turbines introduces different lightning threats in winter and in summer (Méndez et al., 2018; IEC 61400-24, 2019). Using the maps from March et al. (2016), the environmental factor in the standard now includes the local threat of winter lightning. While considering of winter lightning is a good first step, the quality of the risk assessment could be improved because the maps are very coarse, underestimate winter lightning, and use a static threshold ($< 5\,°C$ at 900 hPa). The reasons for the insufficient consideration of lightning in winter in the standard are the different processes leading to upward lightning and the limited meteorological knowledge of lightning in the cold season (Becerra et al., 2018).

> One approach to investigate these differences would be to numerically simulate individual thunderstorms, which may require horizontal resolutions of $\mathcal{O}(100\,\text{m})$ (Bryan et al., 2003) and still fail to make thunderstorms appear at the correct times and places. Our approach takes advantage of already knowing where and when lightning occurs from measurements. Thunderstorm environments can then be identified from reanalysis data that do not need to explicitly resolve thunderstorms or the processes leading to electrification. Morgenstern et al. (2022) used this approach and found that thunderstorms in the cold season differ physically from thunderstorms in the warm season in northern Germany...«

> Line 328: »Stucke et al. (2022) relate our two thunderstorm environments as described in Morgenstern et al. (2022a) to upward lightning at two alpine towers and find that most upward lightning occurs in wind-field thunderstorm conditions. Thus, wind-field thunderstorms pose a particular risk to tall infrastructure and should be considered when determining the lightning threat to wind farms. If Stucke et al. (2022)'s relation between wind-field thunderstorms and upward lightning (i.e., lightning to tall structures) holds also for flat terrain and for the thunderstorm sub-type noMF, then offshore wind farms are at particular risk from wind-field lightning. The lightning protection standard IEC 61400-24 (2019) could be improved by additionally including the proportion of wind-field thunderstorms at sites considered for wind farms. Currently, only the local lightning density, the height of the structure, and an environmental factor (i.e., factors for winter lightning, terrain slope angle, and elevation) are taken into account (IEC 61400-24, 2019). The concept of thunderstorm environments introduced here is superior to the idea of *winter lightning* versus *summer lightning* as it takes into account regional and seasonal differences. The concept is easy transferable to many locations as it is independent of static thresholds as they are used for example by March et al. (2016), Montanyà et al. (2016), or Sherburn and Parker (2014).«

Further future applications of this research are lightning probability maps for each thunderstorm environment to better quantify lightning risk at a given place.

ACTION 2: The use of our work for further research is stated more clearly in the discussion.

> Line 369: »It would also be interesting to model lightning probability maps for each thunderstorm environment in each season, and to investigate the proportion of all thunderstorms in a given region that occur in a particular thunderstorm environment.«

*5) This is accompanied by the most important points of criticism of the submitted manuscript. Both the title and the abstract are misleading, since they give the impression that different types of thunderstorms are actually treated, and moreover, even the differences in lightning physics are discussed. However, to differentiate between thunderstorm types, one would examine smaller scale characteristics, such as subdivision into mesoscale systems or isolated thunderstorm cells.*

Our scope to analyze larger-scale meteorological environments favorable for thunderstorms is stated more explicitly by changing "thunderstorm types" to "thunderstorm environments" throughout the manuscript and to state our objective more clearer and earlier.

ACTION 1: Changing 'thunderstorm types´ to 'thunderstorm environments´.

Rephrasing as a follow-up of the renaming of thunderstorm types into thunderstorm environments.

> Line 386: »Cluster analysis is performed individually on 12 domains in Europe to find and describe different thunderstorm environments and to name them according to their characteristics compared to other thunderstorms in that region. A decision tree is developed to easily distinguish the thunderstorm environments (Fig. 4).«

> Figure 4 (caption): »Decision tree to label the clusters to the corresponding two major thunderstorm environments and the three sub-environments (colored boxes).«

ACTION 3: The abstract is changed to communicate our objective early and clearly:

> Line 1 (abstract): »Meteorological environments favorable for thunderstorms are studied across Europe, including rare thunderstorm conditions from seasons with climatologically few thunderstorms.«

> Line 8 (abstract): »Based on these results it is possible to differentiate lightning conditions in different seasons from coarse reanalysis data without a static threshold or a seasonal criterion.«

*6) Also, the vulnerability of high infrastructure as a motivation and the dissimilarity of winter and summer thunderstorms in terms of their lightning characteristics is not discussed further, except that it is mentioned prominently and is dismissed as "beyond the scope of this paper" in the course.*

ACTION: Tall infrastructures are no longer mentioned in the introduction. A new sentence in the motivation leads the reader to the potential benefits of our research for the protection of tall infrastructures (same issue as referee 1, number 1).

> Line 1 (abstract): »Meteorological environments favorable for thunderstorms are studied across Europe, including rare thunderstorm conditions from seasons with climatologically few thunderstorms.«

> Line 15: »Thunderstorms during the cold season are generally rare but pose a serious threat to wind turbines and other tall structures because it has been observed that lightning strikes to tall infrastructure have no or only a weak annual cycle whereas lightning in general has a pronounced annual cycle (Stucke et al., 2022; Matsui et al., 2020; Vogel et al., 2016).«

*7) This short-coming is already noticeable in the introduction, when there is a strong break in the content between general thunderstorm climatology and the forced-seeming motivation for this paper in lines 40-44. In this respect, more depth may be added to the text.*

ACTION 1: The introduction is restructured for a smoother reading experience. The sentences leading to the strong break are altered and moved further down. More details are provided in the diff-file.

> Line 50: »[...] In general, the European lightning patterns are well described (e.g., Wapler and James, 2015; Enno et al., 2014), but the meteorological drivers leading to lightning in the winter compared to summer are less understood. High structures such as wind turbines or radio towers increase the occurrence of lightning (March et al., 2016), especially in the cold season (Vogel et al., 2016; Pineda et al., 2018) so that

lightning damage to infrastructure is evenly distributed over the year even though lightning occurrence in the surroundings has a strong annual cycle (Stucke et al., 2022).«

ACTION 2: More depth is added to the introduction by motivating our work with the shortcomings in the current lightning protection standard.

See answer 4.

*8) In this sense, the paper can disappoint the reader. Consideration should at least be given to changing the title, abstract, and also the summary accordingly to match the results of the paper. These are essentially described above and can be understood as an indication that thunderstorms are not exclusively caused by strong seasonal heating of the subsurface (solid or liquid), but also independent dynamic factors, called "wind-field" in this paper, play a role.*

Thanks for pointing out the misleading textual formulations in the manuscript. The title, the abstract, and the summary in the discussion section have been changed as indicated above.

ACTION: The conclusions section is enhanced by a more detailed motivation.

Line 373: »This study investigates seasonal and regional differences of meteorological environments in which lightning occur in Europe. Highly destructive lightning damages often occur in seasons and regions where lightning is climatologically unlikely. They pose a challenge for lightning risk assessments because time series of lightning observations are often short and the meteorological conditions for lightning in the cold season are not well understood. This study explicitly includes infrequent lightning conditions by considering an equal number of lightning observations from each season. EUCLID lightning data are combined with meteorological ERA5 data to answer two research questions: "Are there regions in Europe, where thunderstorms occur under similar meteorological conditions?" and "What characterizes thunderstorms in different meteorological environments and how do they vary seasonally across Europe?" Using coarse but consistent reanalysis data, this study paves the way for lightning reconstructions by providing tools to diagnose favorable lightning conditions.«

Line 407: »The process-oriented view of different thunderstorm environments challenges the traditional idea of *winter lightning* versus *summer lightning* and makes it easier to compare similar processes with different magnitudes in different regions.«

*9) Despite these short-comings, the overall quality of the paper is good.*

Thank you.

**Editors' comment**

*Johannes Dahl: I would like to add a comment: The nomenclature of "wind-field" thunderstorms vs. "mass field" thunderstorms is potentially misleading I think. To some readers, it might suggest that wind-field storms are somehow driven by the wind field, rather than by CAPE (because you contrast them with "mass-field" thunderstorms). Really, what you are saying is that in some regimes, CAPE is a poor predictor for whether or not lightning occurs. But this appears to be a result of CAPE being generally small in these regimes (and hence, it has only small variability). Maybe you could add some clarification in the revised manuscript.*

What we are saying is that there are two thunderstorm environments for which the charge separation process differs. The release of CAPE in mass-field thunderstorms causes a quasi-vertical charge separation. On the other hand, high wind speeds and shear separate charged particles along a slanted path in wind-field thunderstorm environments without the need for any release of CAPE. Consequently, CAPE is a poor predictor and cannot explain the development of these thunderstorms. We show that these non-CAPE thunderstorms occur all over Europe, mostly in the cold season, and that they originate in a variety of weather patterns.

ACTION 1: More emphasis is placed on the reduced CAPE values in wind-field thunderstorms.

> Line 3 (abstract): »Wind-field thunderstorms, characterized by increased wind speeds, high shear, strong large-scale vertical velocities, and low CAPE, occurring mainly in winter;«

> Line 400: »In this environment, CAPE is a poor predictor of whether lightning will occur because it is generally small. High wind speeds and shear cause the charged particles to be separated along slanted paths. Wind-field thunderstorms dominate ... «

> Line 248: »CAPE is thus a poor predictor and cannot explain the development of these thunderstorms.«

ACTION 2: The differences between the two major thunderstorm environments are made clearer by discussing the different mechanisms of charge separation.

> Line 249: »Charged particles in wind-field environments are therefore not separated in a quasi-vertical direction through the release of CAPE but instead along a slanted path driven by high horizontal wind speeds combined with shear and strong large-scale vertical velocities - hence the label 'wind-field´ thunderstorm environment.

> The reddish mass-field thunderstorm environment in all domains have high mass-field variables with large CAPE, high water vapor concentrations, and elevated -10 °C isotherm heights. Consequently, charge separation occurs quasi-vertically when CAPE is released. «

> Line 322: »Because high shear results in tilted clouds, charge separation in wind-field thunderstorm environments occurs along a slanted path within the cloud, resulting in charge centers that are also horizontally separated. This is known as the *tilted charge hypothesis* (Takeuti et al., 1978; Brook et al., 1982; Engholm et al., 1990; Williams, 2018; Takahashi et al., 2019; Wang et al., 2021), and is often described for cold season thunderstorms. Those thunderstorms do not require the release of CAPE to explain charge separation, so CAPE is often low and consequently a poor predictor of such thunderstorms.«

> Line 216: For better readability the reference to Market et al., 2002 is moved further up in the discussion.

> Line 393: »The release of CAPE results in a quasi-vertical separation of the charged particles. Mass-field thunderstorm environments occur mostly in the warmer seasons ...«

**Further changes**

**The term strong updrafts is potentially misleading in the context of wind-field thunderstorms. Updrafts commonly refer to a small region of updrafts in convective systems. Updrafts in the context of wind-field thunderstorms actually refer to high large-scale vertical wind speeds.**

The variable refers to the maximum upward vertical velocity within the ERA5 grid cell and one hour. To prevent confusion all instances are renamed to large-scale vertical velocity.

ACTION: Renaming updrafts to large-scale vertical velocity.

**L. 48: The reference Holle et al., 2016 referred to the wrong paper. Holle wrote more than one paper in 2016.**

ACTION: Holle et al., 2016 refers now to the correct paper (see references).

**Online supplement.**

ACTION: The online supplement is changed according to the changes described in this letter. The new doi is: 10.5281/zenodo.7695803

**Style improvements**

**1) Replacing words.**

a) 'Provides' instead of 'gives'.

> Line 19: »A comparison between different regions provides a comprehensive overview of the lightning characteristics in Europe.«

b) 'Secondary maximum' instead of 'secondary peak'.

> Line 26: »...with a maximum over land in summer (MJJA) and a secondary peak in fall and early winter (SONDJ) ...«

c) Replacing 'if' by 'whether'. Replacing 'leading to' by 'resulting in'. Replacing 'in Europe' by 'across Europe'.

> Line 33: »The question remains whether there are meteorologically different thunderstorm conditions at work, resulting in these spatial and temporal differences in lightning characteristics across Europe.«

d) 'Presented' instead of 'compared'.

> Line 76: »The general thunderstorm conditions in Europe are presented in Sect. 4.1 ...«

e) 'longitude/latitude' instead on 'lon/lat'.

> Figure 1 (caption): »Topographic data is based on ERA5 orography with a resolution of $0.25° \times 0.25°$ longitude/latitude.«

f) 'Within' instead of 'from'.

> Line 138: »... to capture everything within the hour in which lightning was observed.«

g) 'An already existing' instead of 'the same'.

> Line 263: »With $k > 3$ more and more clusters are found referring to an already existing thunderstorm environment revealing no additional meteorological insights.«

h) 'Clearly separate' instead of 'are clearly separable from one another'.

Line 300: »First of all, the figure shows that the two major thunderstorm environments [...] clearly separate as the bluish and reddish circles are located in different parts of the figure.«

i) 'Marine' instead of 'maritime'.

Line 352: »Large parts of the UK and Ireland are classified as marine regions«

j) 'Tall' instead of 'high'.

Line 365: »How often are wind turbines or other tall structures struck by which thunderstorm environment?«

k) 'Divided' instead of 'separated'. 'where' instead of 'in which'.

Line 382: »Using principal component analysis, the European territory can be divided into four regions where the atmospheric conditions for thunderstorms are similar throughout the year.«

l) 'Concentrations' instead of 'masses'.

Line 383: »The alpine-central region with thick clouds, large cloud particle concentrations, ...«

m) 'Domain' instead of 'region'.

Line 387: »... and to name them according to their characteristics compared to other thunderstorms in that region domain.«

n) Hyphens instead of commas.

Line 390: »There are two major thunderstorm environments – wind-field thunderstorms and mass-field thunderstorms – and three sub-environments thereof.«

o) 'Enhanced' instead of 'increased'. 'Amounts of precipitation' instead of 'precipitation amounts'. 'Often occurs' instead of 'occurs often'.

Line 403: »Sometimes the cloud-physics (CP) variables are additionally enhanced leading to the wind-field CP thunderstorm sub-type with large cloud sizes, increased concentrations of cloud particles (snow, ice, supercooled liquids), and large amounts of precipitation. Another sub-type, wind-field noMF is characterized by decreased mass-field variables (no MF) and often occurs over the sea.«

*2) Rephrasing to be more concise.*

a) Rephrased.

Line 39: »The most pronounced diurnal cycle is found over mountainous areas and commonly explained by the topography. Complex terrain favors ...«

Line 69: »Here, their findings are extended to large parts of Europe to answer two questions.«

Line 105: »*Topographic variables* consist of the ...«

Line 172: »Thunderstorms in 12 European domains ...«

Line 295: »..., and the question remains how similar the thunderstorm environments of the same name from different domains are.«

Line 366: »What is the relationship between thunderstorm environments and lightning properties ...«

Line 397: »The other major thunderstorm environment, wind-field thunderstorms, originate in more diverse weather conditions, …«

b) Instead of 'using principal component analysis…':

Line 7 (abstract): »Principal component analysis is used to identify four topographically distinct regions in Europe that share similar thunderstorm characteristics«

c) Instead of 'Lightning may originate in various meteorological settings…':

Line 12: »Lightning, the defining characteristic of thunderstorms, can originate in a variety of meteorological settings. Some conditions that lead to lightning…«

**3) Omitting unnecessary parts of a sentence.**

Line 31: »In the northern Atlantic region, occasional intense thunderstorms are possible, even though the climatological thunderstorm activity is low (Enno et al., 2020).«

Line 74: »To answer these questions, Europe is divided into 12 domains (Sect. 2).«

Line 84: »Meteorological data are extracted from the single-level and model-level data of the ERA5 global reanalysis provided by ECMWF.«

Line 101: »All 25 variables are listed in Table 1 and details about them are provided in the online supplement.«

Line 123: »…a basin surrounded by mountain ranges.«

**4) Adding an example.**

Line 44: »but the mountain ranges in southern Europe have thunderstorm frequencies of > 60 thunderstorm days per year (e.g., northern Italy).«

**5) When the names of the thunderstorm environments are introduced, they are set to italics (Sect. 4.2).**

---

## Author Response (AR2)

**Editors' second comment**

*Johannes Dahl:*

**The remaining reviewer recommending accepting the paper as is, but I am going to render a decision of minor revisions, as I think additional clarifications are required.**

**I appreciate your explanations regarding the designation of wind-field vs. mass field thunderstorms. However, I think there is still a misunderstanding, which needs to be clarified before the paper can be accepted for publication. Based on your replies, you state that wind field thunderstorms "... do not require the release of CAPE..." (line 322) and that "CAPE ... cannot explain the development of these thunderstorms" (line 248). I am not aware of any recent study that demonstrated that deep convection can occur in the absence of any CAPE. Some of the cold-season storms you sampled likely occurred within deeply mixed polar air masses, or in association with narrow cold-frontal rainbands. It seems possible that the dataset you used doesn't sample these small CAPE values very well (especially the narrow sliver of weak CAPE ahead of narrow cold frontal rainbands), but I am uncomfortable with the statements that no CAPE is required at all with these storms (I checked the Takahashi et al. 2019 reference, and they merely state that CAPE was weak, rather than absent). As you are certainly aware, the mere presence of a sheared flow is not sufficient for deep convection to develop. You generally need just a little CAPE to get deep convection going, but the strength and organization of the resulting storms is indeed strongly modulated by the vertical wind shear; so, it stands to reason that shear ends up being a reasonable predictor for the intensity of the convection (conditional on the presence of convection, which requires CAPE). The CAPE ingredient for deep convection has been extremely well tested over decades of convective forecasting around the world, so I am a little surprised at the claim that a large fraction of storms supposedly develops without any instability being present. I thus maintain that from a physical perspective the distinction between "mass-field" and "wind-field" thunderstorms has little merit (all deep convection is tied to a vertical mass-field distribution that allows for conditional instability). Once this issue has been clarified, I'm ready accept the paper.**

Thanks for insisting on having the issue properly clarified! Indeed, CAPE is required, and we should better write that CAPE is *low* in wind-field thunderstorms instead of *absent*. Our motivation for the name *wind-field thunderstorm environment* is that these thunderstorms, in addition to having low mass-field values, also have high horizontal wind-speeds, causing charge separation along a tilted path. The low (but present) CAPE values alone would probably not be sufficient to produce lightning if the wind-field variables were also low.

Based on your feedback, the following changes are made:

ACTION 1: The sentences mentioned (l. 322 and l. 248) are either omitted or rephrased. CAPE is neither 'absent' nor 'not required'. Instead of 'decreased wind-field values', there are now 'low wind-field values compared to other thunderstorms'.

ACTION 2: It is stated more clearly, that wind-field thunderstorms do have increased mass-field values compared to situations without lightning, but the values are small compared to the values in mass-field environments. To provide a clearer context for our statements, we state more clearly that the described thunderstom environments are valid relative to other thunderstorm conditions.

> Line 3 (abstract): »Wind-field thunderstorms, characterized by increased wind speeds, high shear, strong large-scale vertical velocities, and low CAPE values compared to other thunderstorms in the same region; and mass-field thunderstorms, characterized by large CAPE values, high dew point temperatures, and elevated isotherm heights. Wind-field thunderstorms occur mainly in winter and more over the seas while mass-field thunderstorms occur more frequently in summer and over the European mainland.«

> Line 225: »*Wind-field thunderstorms* are characterized by increased wind-field values and rather low mass-field variables compared to other thunderstorm conditions in the same domain, and are indicated by bluish colors. There are two wind-field sub-environments: *Wind-field$_{CP}$ thunderstorms* (dark blue) that have additionally enhanced cloud-physics variables (CP), and *wind-field$_{lowMF}$ thunderstorms* (light blue), where the dominant feature is particularly low mass-field values compared to other thunderstorms (low MF), while wind-field variables and cloud-physics variables are at their average values. Compared to conditions without lightning, the moisture and temperature profiles (mass-field values) are still increased in wind-field environments (Morgenstern et al., 2022).

The other major thunderstorm environment, *mass-field thunderstorms*, is characterized by average or increased mass-field variables compared to the wind-field thunderstorms plus often decreased surface-exchange variables and is indicated by reddish colors.«

Captions Fig. 4 + 5: »CP = additionally increased cloud-physics variables, lowMF = decreased mass-field variables, and SX = increased surface-exchange variables compared to other thunderstorms in the same domain.«

Line 246: »and large boundary layer heights of more than 1200 m (Table 2) compared to other thunderstorms in domain B.«

Line 254: Omitting "CAPE ... cannot explain the development ..., Instead: »The common feature of these thunderstorms is increased horizontal wind speeds - hence the label 'wind-field´ thunderstorm environment.«

Line 256: »The reddish mass-field thunderstorm environment has in all domains high mass-field variables with large CAPE values, high water vapor concentrations, and elevated -10 °C isotherm heights compared to other thunderstorms in the same domain.«

Line 317: »Their investigations revealed the importance of vertical temperature difference between the surface and mid-troposphere (700/500 hPa) and low-tropospheric wind shear, while CAPE was no useful predictor.«

Line 326: »this HSLC concept relates to our wind-field thunderstorm environments, especially the sub-environment with particularly low mass-field values (wind-field$_{lowMF}$).«

Line 408: »...but share the characteristics of average or reduced low values in mass-field variables«

Line 416: » Another sub-environment, wind-field$_{lowMF}$, is characterized by decreased particularly low mass-field variables (no low MF) and often occurs over the sea.«

ACTION 3: The thunderstorm sub-environment *wind-field$_{noMF}$* is renamed to *wind-field$_{lowMF}$* to emphasize that the mass field values are present, but low.

ACTION 4: Increased wind-field values are no longer the 'driver' for wind-field thunderstorms but 'occur together'.

Line 245: »Figure 5 shows, that the wind-field thunderstorms (middle-blue triangles) in domain B occur together with enhanced wind speeds, enhanced...«

ACTION 5: The link to the tilted charge hypothesis is now first made in the discussion. The low CAPE values in this context are supported by numbers.

Line 245: Instead of the link to the tilted charge hypothesis in the results section: »The common feature of these thunderstorms is increased horizontal wind speeds - hence the label 'wind-field´ thunderstorm environment.«

Line 328: »High shear, as it is characteristic of wind-field thunderstorms, results in tilted clouds. Thus, charge separation occurs along a slanted path within the cloud, and the charge centers are also separated horizontally. This is known as the *tilted charge hypothesis* [...], and is often described for cold season thunderstorms. We think that this tilt provides sufficiently large distances between the charge centers to cause lightning, even though the clouds are shallow and CAPE is low, with cluser-means of approximately 100 J kg$^{-1}$ or less (Table **??**). In mass-field thunderstorms, on the other hand, CAPE often exceeds 300 J kg$^{-1}$ (factor 3 and more), and horizontal wind speeds are lower, so that charge can separate along a more upright path. Both types require conditionally unstable parts in the temperature and moisture profiles, and thus have increased mass-field values compared to conditions without lightning (Morgenstern et al., 2022). Thunderstorms with tilted clouds are referred to as *wind-field* thunderstorms to emphasize that lightning is unlikely to occur when, in addition to low CAPE values, the horizontal wind speeds are also low.«

ACTION 6: Grammar, spelling, and punctuation are improved, and some rewriting is done without changing the meaning of the text. These changes are best tracked in the diff-file.

**Editors' first comment**

*Johannes Dahl: I would like to add a comment: The nomenclature of "wind-field" thunderstorms vs. "mass field" thunderstorms is potentially misleading I think. To some readers, it might suggest that wind-field storms are somehow driven by the wind field, rather than by CAPE (because you contrast them with "mass-field" thunderstorms). Really, what you are saying is that in some regimes, CAPE is a poor predictor for whether or not lightning occurs. But this appears to be a result of CAPE being generally small in these regimes (and hence, it has only small variability). Maybe you could add some clarification in the revised manuscript.*

What we are saying is that there are two thunderstorm environments for which the charge separation process differs. The release of CAPE in mass-field thunderstorms causes a quasi-vertical charge separation. On the other hand, high wind speeds and shear separate charged particles along a slanted path in wind-field thunderstorm environments without the need for any release of CAPE. Consequently, CAPE is a poor predictor and cannot explain the development of these thunderstorms. We show that these non-CAPE thunderstorms occur all over Europe, mostly in the cold season, and that they originate in a variety of weather patterns.

ACTION 1: More emphasis is placed on the reduced CAPE values in wind-field thunderstorms.

> Line 3 (abstract): »Wind-field thunderstorms, characterized by increased wind speeds, high shear, strong large-scale vertical velocities, and low CAPE, occurring mainly in winter;«

> Line 400: »In this environment, CAPE is a poor predictor of whether lightning will occur because it is generally small. High wind speeds and shear cause the charged particles to be separated along slanted paths. Wind-field thunderstorms dominate ... «

> Line 248: »CAPE is thus a poor predictor and cannot explain the development of these thunderstorms.«

ACTION 2: The differences between the two major thunderstorm environments are made clearer by discussing the different mechanisms of charge separation.

> Line 249: »Charged particles in wind-field environments are therefore not separated in a quasi-vertical direction through the release of CAPE but instead along a slanted path driven by high horizontal wind speeds combined with shear and strong large-scale vertical velocities - hence the label 'wind-field´ thunderstorm environment.

> The reddish mass-field thunderstorm environment in all domains have high mass-field variables with large CAPE, high water vapor concentrations, and elevated -10 °C isotherm heights. Consequently, charge separation occurs quasi-vertically when CAPE is released. «

> Line 322: »Because high shear results in tilted clouds, charge separation in wind-field thunderstorm environments occurs along a slanted path within the cloud, resulting in charge centers that are also horizontally separated. This is known as the *tilted charge hypothesis* (Takeuti et al., 1978; Brook et al., 1982; Engholm et al., 1990; Williams, 2018; Takahashi et al., 2019; Wang et al., 2021), and is often described for cold season thunderstorms. Those thunderstorms do not require the release of CAPE to explain charge separation, so CAPE is often low and consequently a poor predictor of such thunderstorms.«

> Line 216: For better readability the reference to Market et al., 2002 is moved further up in the discussion.

> Line 393: »The release of CAPE results in a quasi-vertical separation of the charged particles. Mass-field thunderstorm environments occur mostly in the warmer seasons ...«